# *WNT10A* mutation causes ectodermal dysplasia by impairing progenitor cell proliferation and KLF4-mediated differentiation

Mingang Xu[1], Jeremy Horrell[1], Melinda Snitow[2], Jiawei Cui[1], Heather Gochnauer[1], Camille M. Syrett[1], Staci Kallish[2], John T. Seykora[1], Fei Liu[3], Dany Gaillard[4], Jonathan P. Katz[2], Klaus H. Kaestner[5], Brooke Levin[6], Corinne Mansfield[7], Jennifer E. Douglas[7], Beverly J. Cowart[7], Michael Tordoff[7], Fang Liu[1], Xuming Zhu[1], Linda A. Barlow[4], Adam I. Rubin[1], John A. McGrath[8], Edward E. Morrisey[2], Emily Y. Chu[1] & Sarah E. Millar[1,9]

Human *WNT10A* mutations are associated with developmental tooth abnormalities and adolescent onset of a broad range of ectodermal defects. Here we show that β-catenin pathway activity and adult epithelial progenitor proliferation are reduced in the absence of WNT10A, and identify Wnt-active self-renewing stem cells in affected tissues including hair follicles, sebaceous glands, taste buds, nails and sweat ducts. Human and mouse *WNT10A* mutant palmoplantar and tongue epithelia also display specific differentiation defects that are mimicked by loss of the transcription factor KLF4. We find that β-catenin interacts directly with region-specific LEF/TCF factors, and with KLF4 in differentiating, but not proliferating, cells to promote expression of specialized keratins required for normal tissue structure and integrity. Our data identify WNT10A as a critical ligand controlling adult epithelial proliferation and region-specific differentiation, and suggest downstream β-catenin pathway activation as a potential approach to ameliorate regenerative defects in *WNT10A* patients.

[1] Department of Dermatology, Perelman School of Medicine, University of Pennsylvania, Philadelphia, Pennsylvania 19104, USA. [2] Department of Medicine, Perelman School of Medicine, University of Pennsylvania, Philadelphia, Pennsylvania 19104, USA. [3] Molecular & Cellular Medicine Department, Texas A&M University Health Science Center, College Station, Texas 77843, USA. [4] Department of Cell and Developmental Biology, and Rocky Mountain Taste and Smell Center, University of Colorado School of Medicine, Aurora, Colorado 80045, USA. [5] Department of Genetics, Perelman School of Medicine, University of Pennsylvania, Philadelphia, Pennsylvania 19104, USA. [6] William G. Rohrer Cancer Genetics Program, M.D. Anderson Cancer Center at Cooper, Camden, New Jersey 08103, USA. [7] Monell Chemical Senses Center, Philadelphia, Pennsylvania 19104, USA. [8] Department of Medical and Molecular Genetics, St John's Institute of Dermatology, King's College London, London SE1 9RT, UK. [9] Department of Cell and Developmental Biology, Perelman School of Medicine, University of Pennsylvania, Philadelphia, Pennsylvania 19104, USA. Correspondence and requests for materials should be addressed to S.E.M. (email: millars@mail.med.upenn.edu).

**W**NT10A is the most commonly mutated gene in human non-syndromic selective agenesis of permanent teeth[1,2] and WNT10A mutations are also associated with the ectodermal dysplasia syndromes Odonto-onycho-dermal dysplasia (OMIM #257980) and Schöpf–Schulz–Passarge syndrome (OMIM #224750)[3]. Patients with WNT10A mutations exhibit variable developmental dental defects including microdontia of primary teeth, defective root and molar cusp formation, and complete absence of secondary dentition[2,3]. Non-dental anomalies, such as palmoplantar keratoderma, thinning hair, sweating abnormalities, a smooth tongue surface and defective nail growth, appear beginning in adolescence or even later[4,5], suggesting possible roles for WNT10A in epithelial regeneration. In line with this, genome-wide association studies identified an association between a WNT10A intronic single-nucleotide polymorphism (SNP) that correlates with lower WNT10A mRNA levels, and male pattern baldness[6]. Delineating the basis for these phenotypes and the molecular mechanisms of WNT10A action will be crucial in understanding the developmental and regenerative functions of WNT10A, and designing potential therapeutic approaches for affected individuals.

Here we describe a new human pedigree carrying a predicted loss-of-function mutation in WNT10A and delineate the functions and mechanisms of WNT10A signalling in dental development and adult epithelial renewal by analysing human patient tissue and loss-of-function mouse mutants. We demonstrate that Wnt-activated self-renewing stem cells are present in the adult tissues affected by WNT10A mutation, and identify WNT10A/β-catenin signalling as a broadly used mechanism controlling epithelial progenitor proliferation. In addition to proliferative defects, we unexpectedly identified a requirement for WNT10A/β-catenin signalling in permitting regionally restricted, suprabasal differentiation programmes in tongue filiform papillae and palmoplantar epidermis, explaining the smooth tongue and palmoplantar keratoderma phenotypes observed in human patients. We show that in differentiating suprabasal cells, but not basal progenitor cells, β-catenin complexes with KLF4, a suprabasally restricted transcription factor required for epidermal differentiation programmes[7,8], allowing β-catenin to switch from pro-proliferative to pro-differentiation modes. Our data further suggest activation of the β-catenin pathway as a potential means for restoring normal epithelial functions in WNT10A patients.

## Results

**Human pedigree with a novel loss-of-function WNT10A mutation.** Here we report a 41-year-old woman of Indian descent who contacted our dermatology clinic complaining of thinning hair (Fig. 1a), onychodystrophy (Fig. 1b), palmoplantar scaling (Fig. 1c,d) and decreased palmoplantar sweating (Fig. 1e,f). The patient's tongue surface was abnormally smooth (Fig. 1g,h). Taste testing did not reveal decreased sensitivity to salt, sweet and bitter tastes (Fig. 1i,j); however, her affective (like versus dislike) taste response was blunted compared with her affective response to odors. Her low ability to taste quinine was concordant with genotyping for a TAS2R19 allele associated with quinine sensitivity (heterozygous A:G for rs10772420)[9] and homozygosity for the non-taster diplotype (AVI/AVI) for bitter receptor gene TAS2R38 (ref. 10). She had low alveolar bone density and a history of severe dental defects including small, conical primary teeth with taurodontism, and complete failure of secondary dentition (Fig. 1k).

Genetic testing ruled out mutation of the ectodermal dysplasia-associated genes EDA, EDAR and EDARRAD[11]. Instead, the patient was homozygous for a single nucleotide G > A transversion at position c.756 + 1 (c.756 + 1 G > A) of WNT10A,

which mutates a conserved mRNA splice donor site for intron 3 (Fig. 1l). The proband's younger brother was homozygous for the same mutation and had similar symptoms including conical primary teeth, failure to develop permanent teeth, alopecia and palmoplantar scaling. Both parents were heterozygous carriers, and a male second cousin on her paternal side also reported dental abnormalities.

qPCR of WNT10A transcripts isolated from the patient's plucked scalp hairs revealed the presence of normally spliced exon 1 and exon 2 transcripts at levels comparable to those detected in a similarly aged control female of Indian descent. However, transcripts resulting from splicing of intron 3 were present at <10% of control levels (Fig. 1m). The predicted translation product is truncated after amino acid 252 (Gln), resulting in absence of 16 of the 24 conserved C-terminal cysteine residues necessary for disulfide bridge formation, Wnt protein secondary structure[12] and binding to Frizzled receptors[13]. As human patients with a wide range of different WNT10A mutations including homozygous missense mutations and mutations predicted to truncate the protein at nine amino acids[3] display overlapping phenotypes, these likely result from loss of WNT10A function.

**Localization of Wnt signalling and Wnt10a expression.** Wnt/β-catenin signalling stabilizes cytoplasmic β-catenin, allowing it to accumulate and enter the nucleus where it associates with TCF/LEF family DNA-binding factors, and activates target gene expression. Wnt/β-catenin signalling is active in embryonic ectodermal appendages and is required for their formation[14]. In adult life, Wnt/β-catenin signalling localizes to interfollicular epidermis, hair follicles (HFs), and tongue filiform papillae and taste buds (TBs)[15–17]. Immunofluorescence detection of nuclear β-catenin, and analysis of Wnt/β-catenin reporter expression in Axin2[lacZ], Axin2-Cre[ERT2/tdT](Axin2-tdT) and TCF/Lef-H2B-GFP (TL-GFP) mice[16,18,19] revealed signalling in additional tissues that show defects in WNT10A patients, including sweat gland germs; adult sweat gland ducts, but not secretory cells; footpad epidermis; and basal and differentiated TB cells in adult fungiform and circumvallate taste papillae (Fig. 2a–h''). In tongue filiform papillae, Axin2-tdT expression localized to basal cells and both anterior HOXC13− and posterior HOXC13+ differentiating cells (Fig. 2i–i''), while TL-GFP was expressed in LEF1+ basal and HOXC13− differentiating cells but not in HOXC13+ cells (Fig. 2j–k''), indicating differential sensitivity of these reporters to Wnt signalling. In addition to Wnt activity in HF matrix, pre-cortical and dermal papilla (DP) cells, we also detected Wnt reporter expression in the HF dermal sheath, isthmus and sebaceous gland peripheral cells (Fig. 2l–n).

Wnt10a expression coincides with β-catenin signalling in molar tooth development[20,21], in embryonic anlagen for HFs and taste papillae, in adult interfollicular epidermis, and in HF epithelial cells and DP[16,22–24]. We detected Wnt10a expression in plantar and footpad epidermis at similar levels to those in haired skin epidermis, and in regenerating adult epithelia including filiform and fungiform papillae and sweat ducts (Fig. 2o–t). Wnt10a localized to sweat gland myoepithelial cells, but was not detectable in sweat gland mesenchyme (Fig. 2p).

**WNT10A/β-catenin signalling in molar tooth development.** To study the functions and mechanisms of action of WNT10A in vivo, we generated conditional Wnt10a[fl/fl] mutant mice with loxP sites flanking exons 3 and 4, which encode 20 of the 24 conserved, functionally required cysteine residues (Fig. 2u–w). Wnt10a[fl/fl] mice were crossed with CMV-Cre, Krt14-Cre or Krt5-rtTA tetO-Cre transgenic mice to generate null, constitutive

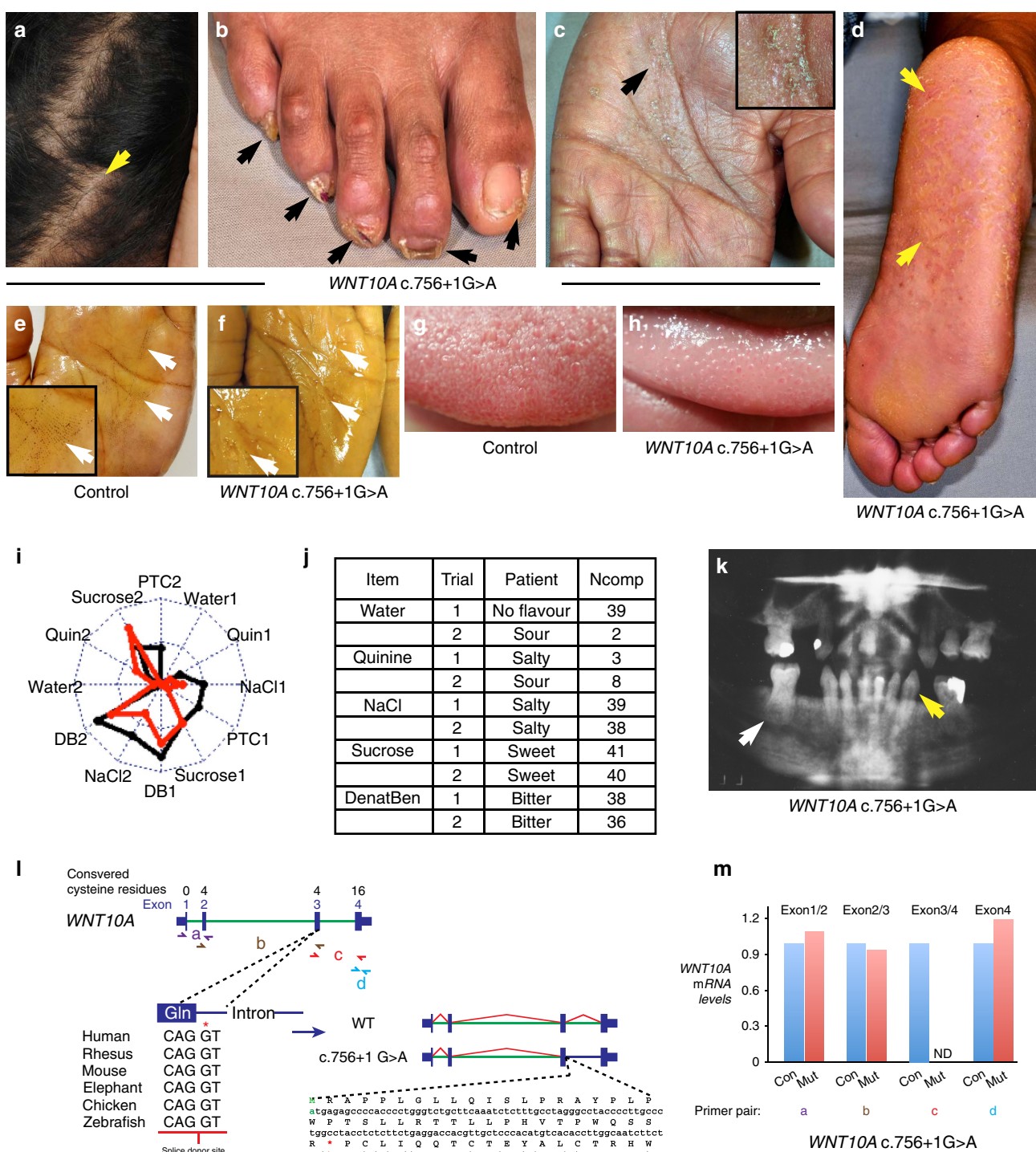

**Figure 1 | Clinical features associated with human *WNT10A* mutation.** (**a**) Thinning hair. (**b**) Nail dystrophy. (**c,d**) Fissures and scaling on palms and soles. (**e,f**) Starch-iodine sweat testing. Note brown grains on control palm indicating sweat production, and decreased sweating in patient (arrows). Insets show higher magnification of areas indicated by the lower arrow in each photograph. (**g,h**) Smooth tongue surface. (**i**) Taste testing. Patient data (red line) is similar to comparison group except for quinine and PTC1 (bitter). DB, denatonium benzoate (bitter). Higher $y$ axis values indicate greater intensity (scale, 1–12). Patient was tested twice; 1 = trial 1; 2 = trial 2. (**j**) Taste quality assessment. NComp, number of comparison subjects ($N_{total} = 41$) who gave same response as patient. (**k**) Oral X-ray. Note decreased alveolar bone density and tooth root and cusp defects. (**l**) Splice donor site mutation and predicted truncated protein. (**m**) qPCR with primers indicated in **l** confirms aberrant splicing.

epithelial, and doxycycline (dox)-inducible epithelial mutants. Approximately 50% of *Wnt10a*$^{-/-}$ mice developed loosely anchored ectopic molar M4 teeth (Supplementary Fig. 1a,b). Maxillary and mandibular molar teeth had flattened cusps, reduced size and defective root bifurcation and extension

compared with littermate controls (Supplementary Fig. 1a,b and Fig. 3a–c), consistent with a previous report[25]. Cusp and size abnormalities were observed by E17.5 (Fig. 3d), indicating their morphogenetic origin, and mimicked the effects of Wnt/β-catenin inhibition after tooth initiation[20]. By 1 year of age,

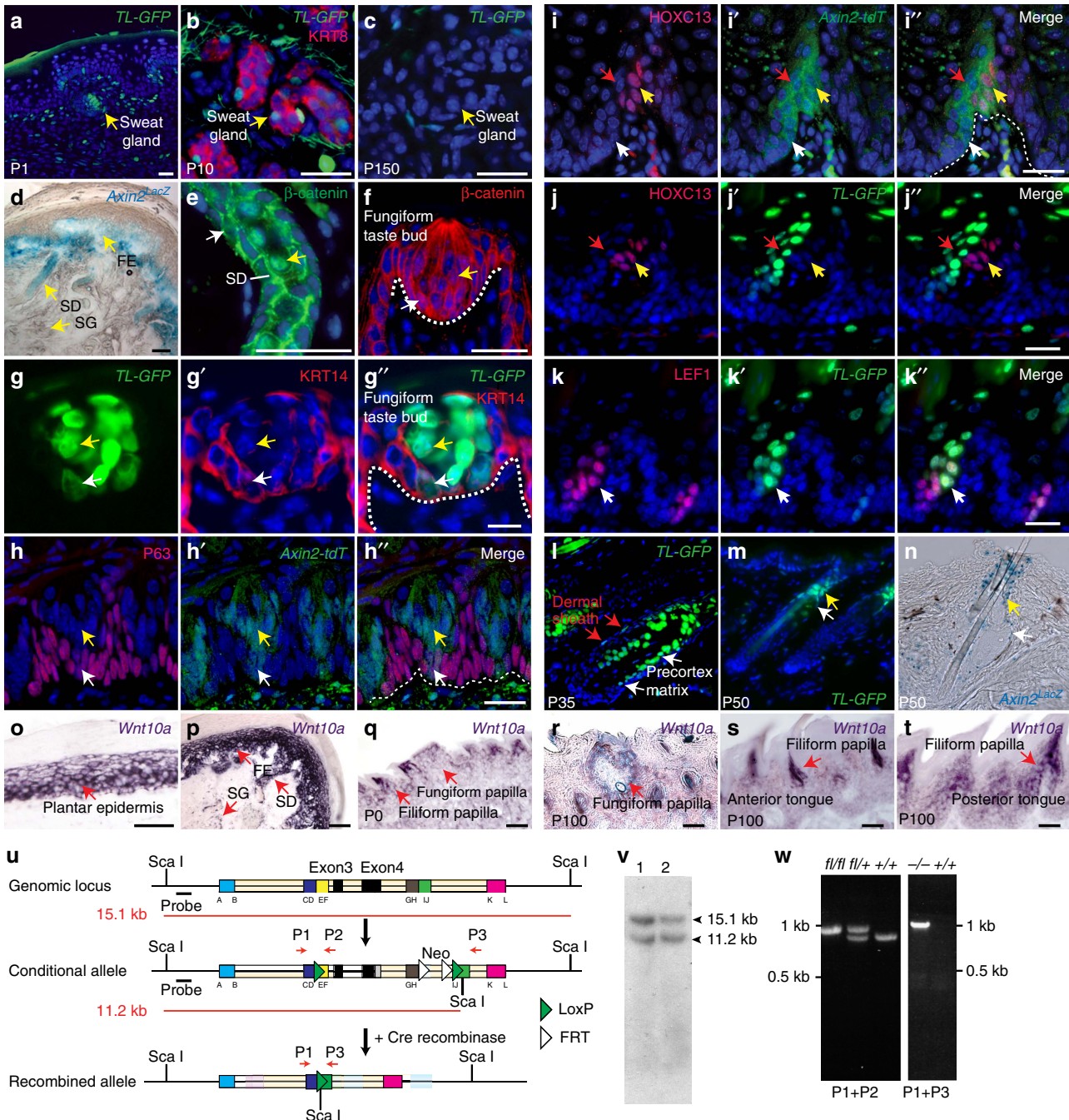

**Figure 2 | Wnt/β-catenin signalling and generation of *Wnt10a* mutant mice.** (**a–c**) *TL-GFP* localizes to sweat gland (SG) germs at P1 (**a**), and KRT8+ SG luminal cells at P10 (**b**), but not adult SG secretory cells (**c**). (**d**) *Axin2^{lacZ}* localizes to sweat ducts (SD) and footpad epidermis (FE), but not SG secretory cells in adult footpad. (**e**) Nuclear and cytoplasmic β-catenin localizes to basal (white arrow) and suprabasal (yellow arrow) cells in adult sweat ducts. (**f**) Nuclear β-catenin localizes to basal (white arrow) and differentiated (yellow arrow) adult fungiform TB cells. (**g–g″**) *TL-GFP* localizes to KRT14+ basal (white arrows) and KRT14− differentiated (yellow arrows) TB cells. (**h–h″**) tdT expression (pink) in p63+ basal (white arrows) and p63− differentiating (yellow arrows) cells in *Axin2-Cre^{ERT2/tdT}* (*Axin2-tdT*) circumvallate papillae. (**i–i″**) tdT expression in posterior HOXC13+ (yellow arrows), anterior HOXC13− differentiating (red arrows) and HOXC13− basal (white arrows) filiform papilla cells in *Axin2-tdT* mice. (**j–k″**) *TL-GFP* localizes to anterior HOXC13− differentiating (**j–j″**, red arrows) and LEF1+ basal (**k–k″**, white arrows) but not posterior HOXC13+ differentiating (**j–j″**, yellow arrows) filiform papilla cells. (**l**) *TL-GFP* localizes to matrix, pre-cortex and dermal sheath in P35 anagen HFs. (**m,n**) *TL-GFP* (**m**) and *Axin2^{lacZ}* (**n**) localize to HF isthmus (yellow arrows) and sebaceous gland peripheral cells (white arrows) at P50 (telogen). (**o–t**) *Wnt10a* expression (*in situ* hybridization, purple) in adult plantar epidermis (**o**); adult footpad epidermis (FE) and sweat ducts (SD) (**p**); neonatal (**q**) and P100 (**r–t**) filiform and fungiform papillae in anterior (**s**) and posterior (**t**) dorsal tongue; and myoepithelial cells of adult SG (lower red arrow) (**p**). (**u–w**) Generation of *Wnt10a*-floxed mice. (**u**) A conditional *Wnt10a* allele was generated by recombination using a cassette with *loxP* sites flanking exons 3–4. Correctly targeted ES cells were confirmed by Southern blotting of Sca1-digested genomic DNA (**v**); probe is indicated in **u**. (**w**) PCR-genotyping with primers P1+P2 confirmed germ-line transmission. *Wnt10a^{fl/fl}* mice were crossed with *CMV-Cre* mice to generate a null allele. PCR-genotyping with primers P1+P3 confirmed deletion of exons 3–4, encoding amino acids 126–528. Primer positions are indicated in **u**. Scale bar, 10 μm (**g–g″**) or 25 μm (other panels).

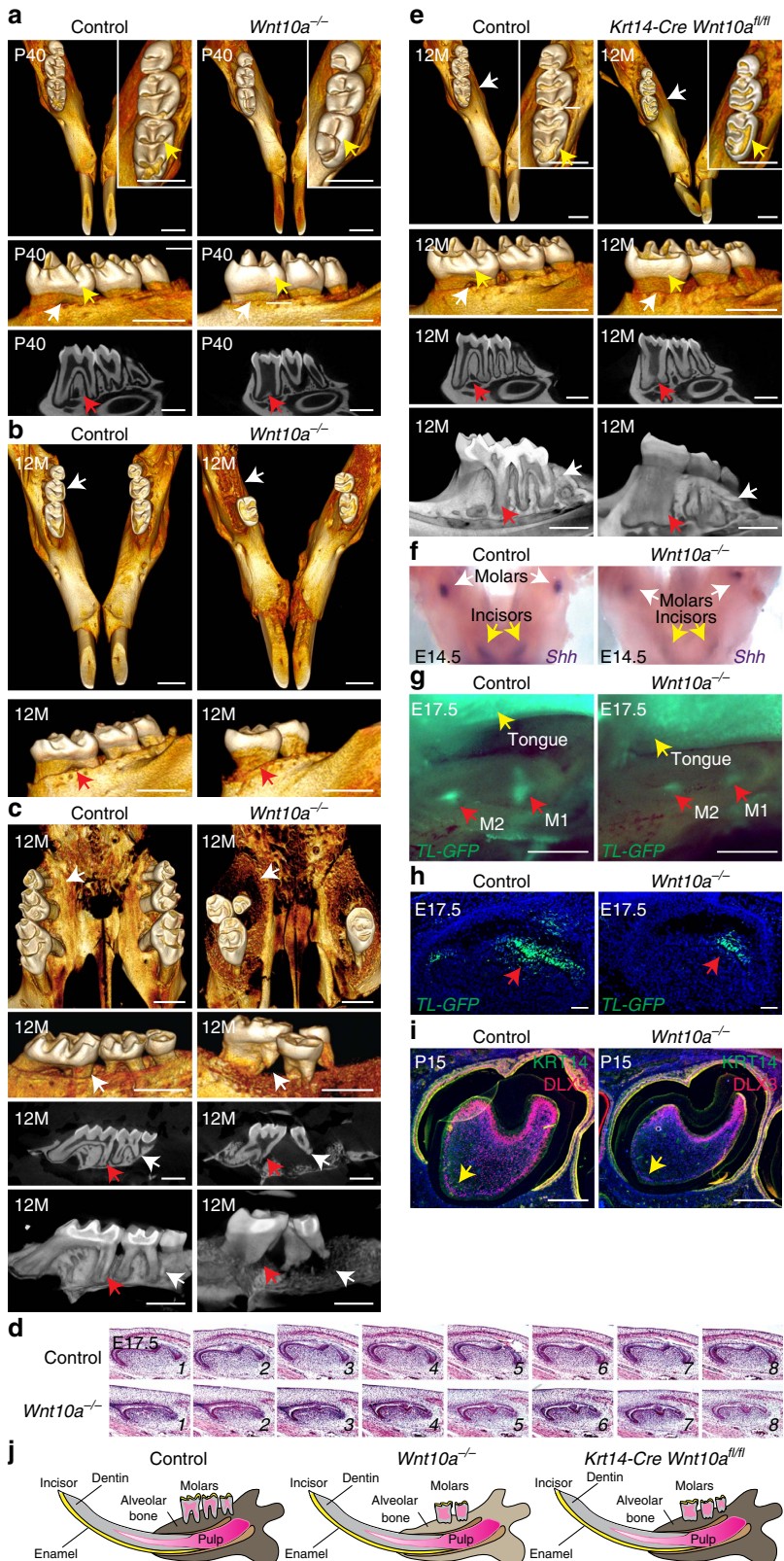

**Figure 3 | Tooth defects in *Wnt10a* mutant mice.** (**a–c**) Micro-CT analysis of mandibles (**a,b**) and maxillae (**c**) from control and *Wnt10a⁻/⁻* mice at the stages indicated. (**d**) Serial H&E-stained sections reveal smaller molar M1 size and blunted cusp development in E17.5 *Wnt10a⁻/⁻* mutant. (**e**) Micro-CT analysis of *Krt14-Cre Wnt10a^{fl/fl}* and control littermate mandibles at 12 months. (**f**) Whole-mount *in situ* hybridization of E14.5 control and *Wnt10a⁻/⁻* mandibles with *Shh* probe. (**g,h**) Analysis of whole mount (**g**) or frozen sections (**h**) of E17.5 *Wnt10a⁻/⁻ TL-GFP* and control *TL-GFP* developing molars shows decreased GFP signal in mutant cusp precursors. (**i**) Immunofluorescence for KRT14 (green) and DLX3 (red) in P15 mandibular molar from control and *Wnt10a⁻/⁻* littermates. *Dlx3* expression is reduced in the mutant. (**j**) Diagram summarizing *Wnt10a* mutant tooth phenotypes at 12 months of age. Scale bar, 50 μm (**h**), 250 μm (**i**) or 1mm (all other panels).

alveolar bone density was markedly decreased in $Wnt10a^{-/-}$ mice compared with controls, in line with human patient phenotypes (Fig. 1k), and molar teeth were frequently missing (Fig. 3b,c). *Krt14-Cre Wnt10a$^{fl/fl}$* mutants displayed molar cusp and root defects, but not decreased bone density (Fig. 3e,j), and a subset of these mice formed ectopic M4 molars. Thus, epithelial *Wnt10a* is required for normal tooth morphogenesis and suppression of ectopic molar formation, and mesenchymal *Wnt10a* maintains alveolar bone. β-catenin signalling in incisor region mesenchyme prevents ectopic incisor development by stimulating expression of *Bmp4* and *Shh*, which then act to limit Wnt activity[26]. *Wnt10a* may be involved in a similar feedback mechanism to suppress ectopic molar formation by signalling to molar mesenchyme. In line with this possibility, $Wnt10a^{-/-}$ mandibles displayed reduced expression of *Shh* at E14.5 (Fig. 3f).

*TL-GFP* expression was reduced in E17.5 $Wnt10a^{-/-}$ mutant molar cusps compared with controls (Fig. 3g,h), indicating that *Wnt10a* regulates cusp formation through the β-catenin pathway. Interactions between Hertwig's epithelial root sheath, a proliferative structure that expresses *Wnt10a* (ref. 21), and adjacent mesenchymal cells control molar root size and shape. The molar root defects in *Wnt10a* mutants phenocopied those caused by deletion of the direct Wnt/β-catenin target gene *Dlx3* in dental mesenchyme[27,28]. DLX3 expression was markedly reduced in mutant root-forming mesenchyme at P15 (Fig. 3i), indicating that *Wnt10a* promotes molar root formation by activating mesenchymal *Dlx3* expression.

### *Wnt10a* in embryonic development of non-dental epithelia.
Primary and secondary HF placodes, taste papilla placodes and sweat gland germs developed normally in *Wnt10a* mutants (Fig. 4a–h). Constitutive epithelial β-catenin deletion in *Krt14-Cre Ctnnb1$^{fl/fl}$* mice caused defective formation of tongue filiform papillae and the tongue barrier, and loss of expression of *Pax9*, a transcriptional factor critical for filiform papilla development and barrier establishment[29] (Fig. 4m–t). By contrast, *Wnt10a* deletion did not grossly affect embryonic development of filiform or fungiform papillae (Fig. 4i–l). Thus, other Wnts may compensate for WNT10A in non-dental embryonic development.

### *Wnt10a* mutant HFs show progressive defects in adult life.
Adult mice with global or constitutive epithelial *Wnt10a* deletion displayed increasingly sparse hair with age. HFs undergo periodic cycles of growth (anagen), regression (catagen) and rest (telogen), driven by rarely proliferating epithelial stem cells in the permanent KRT15+ CD34+ bulge region, and rapidly proliferating progenitors in the adjacent KRT15+ CD34- secondary hair germ (SHG)[30]. Immunofluorescence for pan-hair shaft keratins produced a signal in mutant HFs, and cuticle structure appeared grossly normal by scanning electron microscope (SEM); however, mutant hair shafts were shorter and thinner than controls, with disorganized internal structures (Supplementary Fig. 2a–j). To investigate the mechanisms underlying these defects, we induced epithelial *Wnt10a* deletion at successive time points in *K5-rtTA tetO-Cre Wnt10a$^{fl/fl}$* mice. *Wnt10a* loss from P9 (embryonic anagen) caused premature HF regression, cessation of matrix cell proliferation and decreased cyclin D1 expression (Fig. 5a–b′,j and Supplementary Fig. 2k,l). Deletion at P18 delayed initiation of anagen, indicated by histology and absent SHG proliferation (Fig. 5c–d′), and prevented timely *TL-GFP* activation and external hair growth (Fig. 5k–m). These phenotypes mimicked the effects of Wnt/β-catenin inhibition[16]. Mutant HFs eventually entered anagen by P29 (Fig. 5e,f), but proliferation and cyclin D1 expression remained lower than in controls (Fig. 5e′,f′,i and Supplementary Fig. 2m–p). *Wnt10a* deletion in full postnatal

anagen slightly reduced proliferation but did not cause HF regression (Fig. 5g–i and Supplementary Fig. 2q,r), suggesting compensatory activity of other Wnts. Thus, epithelial WNT10A/β-catenin signalling maintains embryonic anagen and promotes anagen onset.

By 6 months of age, *Wnt10a* mutant HFs became miniaturized with enlarged sebaceous glands and elevated lipid production (Fig. 5n–s). Dominant negative *Lef1* also causes sebaceous gland expansion[31], consistent with decreased Wnt/β-catenin signalling in *Wnt10a* mutant HFs. Despite HF miniaturization in *Wnt10a* mutants, CD34+ KRT15+ bulge stem cells were retained (Fig. 5t–w). Miniaturized HFs in human androgenetic alopecia similarly display enlarged sebaceous glands and bulge stem cell retention[32]. As data from genome-wide association studies indicate association of a *WNT10A* variant with androgenetic alopecia[6], decreased WNT10A/β-catenin signalling may contribute to this condition.

In normal aged mice (18–34 months), HFs miniaturize via loss of stem cells due to COL17A1 proteolysis[33]. Miniaturized HFs of 6-month-old *Wnt10a* mutants expressed COL17A1 (Supplementary Fig. 2s,t), consistent with maintenance of stem cell markers and suggesting that miniaturization was not caused by accelerated aging. In line with this, levels of *Axin2* expression are similar in young and aged HFs[33].

### WNT10A/β-catenin signalling in early postnatal appendages.
Constitutive global or epithelial-specific *Wnt10a* deletion caused progressive defects in tongue papillae structure from ∼P7 (Fig. 6a,b and Supplementary Fig. 3a–b′). TBs were miniaturized, and *TL-GFP* reporter activity was decreased in filiform papillae and TBs compared with controls (Supplementary Fig. 3c–h). Inducible epithelial β-catenin deletion in early postnatal life caused similar defects (Supplementary Fig. 3i,j). Sweat gland ducts failed to extend in $Wnt10a^{-/-}$ mutants (Fig. 6r,s), or following postnatal epithelial β-catenin deletion (Supplementary Fig. 3m,n), and starch-iodine tests revealed a functional inability to sweat (Supplementary Fig. 3k–l′). Thus, WNT10A/β-catenin signalling is required to complete postnatal oral appendage and sweat duct development. In line with this, our patient, and several other human WNT10A pedigrees[5,34], display palmoplantar hypohidrosis. However, palmoplantar hyperhidrosis has also been described in some WNT10A patients[5,35]. We speculate that variable compensatory mechanisms, for instance upregulation of other *Wnt* genes during development, could account for these disparate findings. Approximately 20% of epithelial *Wnt10a* mutants also displayed defects in nail growth (Supplementary Fig. 3o,p), consistent with onychodystrophy in human patients (Fig. 1b).

### WNT10A maintains adult oral appendages and sweat ducts.
To determine whether WNT10A/β-catenin signalling is required for regeneration of adult non-hair bearing epithelia, we examined the effects of inducing epithelial *Wnt10a* or β-catenin deletion, or expression of the Wnt/β-catenin inhibitor DKK1 and its receptor Kremen1, in adult *K5-rtTA tetO-Cre Wnt10a$^{fl/fl}$*, *K5-rtTA tetO-Cre Ctnnb1$^{fl/fl}$* and *K5-rtTA tetO-Dkk1 K14-Krm1* mice[16]. In each case, SEM revealed progressively abnormal filiform and fungiform papilla structures, causing a flattened tongue surface (Fig. 6c,d and Supplementary Fig. 3q,r). SEM or whole-mount immunofluorescence for the TB marker KRT8 revealed decreased TB numbers in mutant fungiform and circumvallate papillae (Fig. 6e–i and Supplementary Fig. 3s). TBs contain Type I, II and III taste receptor cells, marked, respectively, by expression of ENTPD2, PLCβ2 and SNAP-25, and required for glial-like supporting function, and detection of

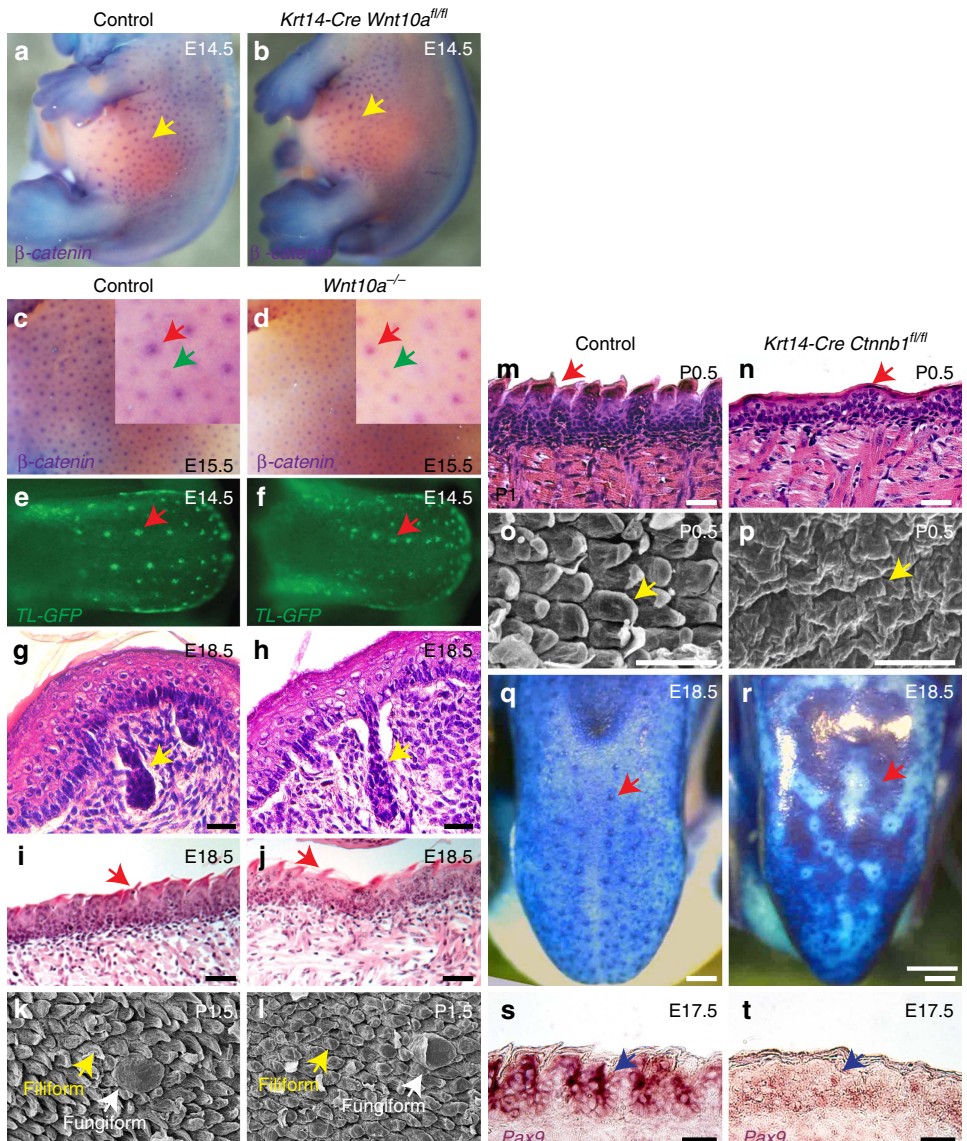

**Figure 4 | *Wnt10a* mutation does not affect embryonic development of HFs or sweat glands or tongue papillae.** (**a–d**) Whole-mount *in situ* hybridization of control and *Wnt10a* mutant embryos at the indicated time points using DIG-labelled probe for β-catenin, an HF placode marker (purple signal), reveals that early stages of HF development are unaffected by constitutive epithelial (**a,b**) or global (**c,d**) deletion of *Wnt10a*. (**e,f**) *TL-GFP* expression in tongue whole mounts at E14.5 shows that fungiform taste papilla placodes and Wnt/β-catenin signalling (arrows) are not affected by global loss of *Wnt10a*. (**g–j**) H&E staining at E18.5 reveals normal initiation of sweat gland (**g,h**, yellow arrows) and filiform papilla (**i,j**, red arrows) development in *Wnt10a* mutants. (**k,l**) P1.5 *Wnt10a* mutant tongues display normal fungiform and filiform papillae, assayed by SEM. (**m–p**) Constitutive epithelial β-catenin deletion causes failure of filiform papilla formation at P0.5: H&E staining (**m,n**); SEM analysis (**o,p**). (**q,r**) Constitutive epithelial β-catenin deletion results in defective tongue barrier formation, revealed by dye penetration (dark blue staining), which is confined to fungiform papillae in controls (**q**, arrow) but is broadly apparent in mutants (**r**, arrow). (**s,t**) *In situ* hybridization for *Pax9* reveals its decreased expression in epithelial β-catenin mutant dorsal tongue (purple/brown signal). $N = 3$ mutants and $n = 3$ control mice were analysed in all experiments. Scale bar, 25 μm (**g,h**), 40 μm (**s,t**), 50 μm (**i,j,m–p**), 100 μm (**k,l**) or 200 μm (**q,r**).

sweet, umami, bitter and sour tastes. While forced β-catenin activation preferentially induces Type I fate[36], *Wnt10a* or β-catenin deletion reduced marker expression for all three cell types (Supplementary Fig. 3s). However, we did not detect significant differences in the taste responses of adult mutants versus controls for sweet, sour, salt or bitter tastes, or for the irritant, capsaicin (Supplementary Fig. 4), indicating that, as in our human patient, residual taste function is sufficient to discriminate these compounds. Interestingly, *Wnt10a*$^{-/-}$ mutants had higher water intakes than controls following mild, but not more severe, water restriction. This could reflect increased dehydration, possibly caused by a slight epidermal barrier defect.

Inducible *Wnt10a* deletion or forced *Dkk1* expression after sweat glands reached maturity at P20 caused sweat duct regression and impaired sweating ability compared with controls (Fig. 6t–w and Supplementary Fig. 3t–u′), revealing a previously unknown role for β-catenin signalling in sweat duct maintenance.

Stratified skin epidermis and tongue epithelium regenerate continuously from keratin 14 (KRT14) + basal cells every 8–10 and 3–4 days, respectively[37,38], while TB cells in fungiform and circumvallate papillae arise from KRT14 + SOX2 + basal cells and have slower turnover rates of up to 3 weeks[39]. Inducible *Wnt10a* deletion in either the early postnatal period, or in adult life, caused significantly decreased basal cell proliferation in

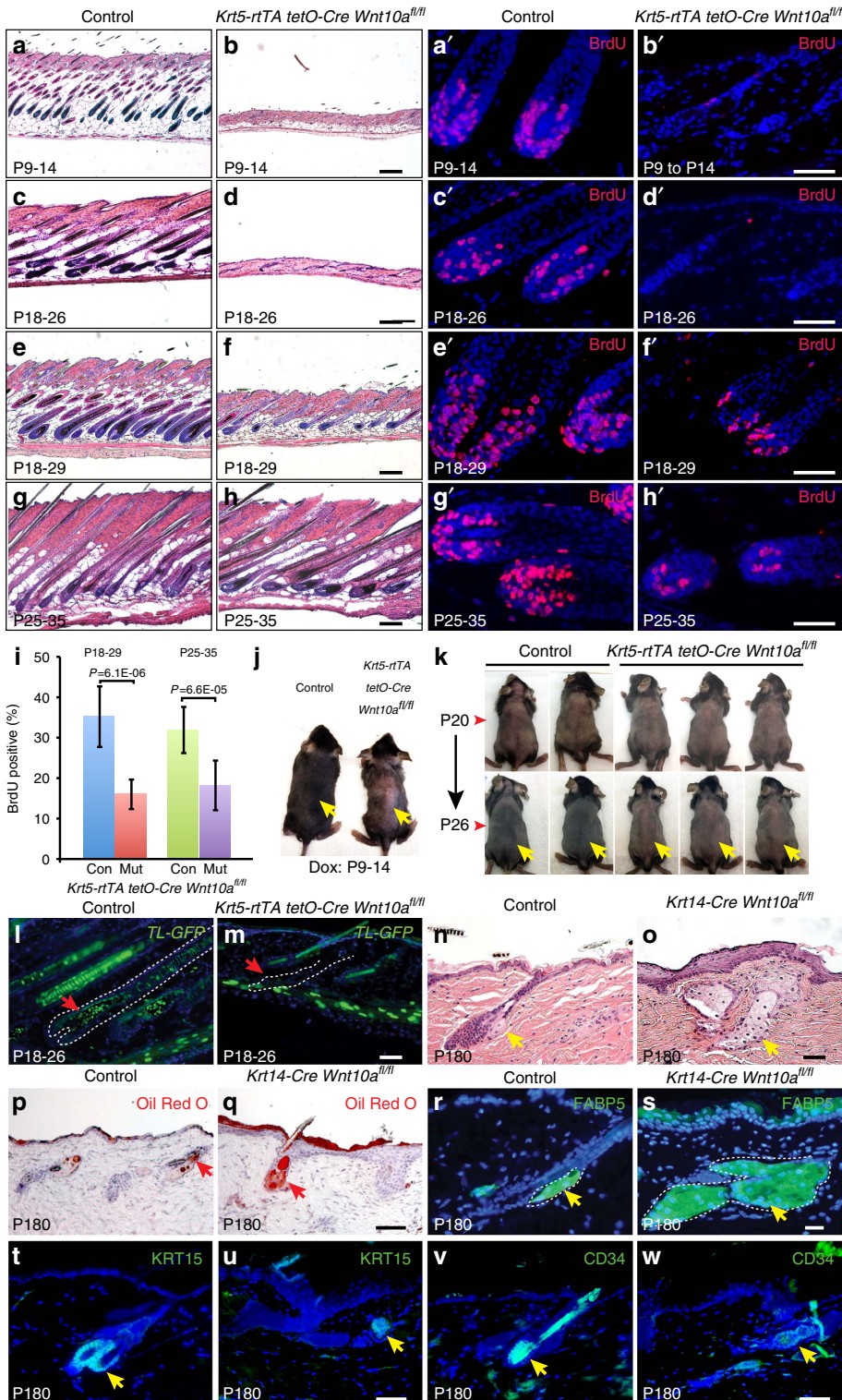

**Figure 5 | *Wnt10a* deletion causes altered hair cycle progression.** (**a–h'**) H&E stained sections (**a–h**) and BrdU immunofluorescence (**a'–h'**) of *Krt5-rtTA tetO-Cre Wnt10<sup>fl/fl</sup>* mutant and littermate control dorsal skin dox induced from P9, P18 or P25 and analysed at the stages indicated (**a–b'**, n = 3 controls, 3 mutants; **c–d'**, n = 6 controls, 6 mutants; **e–f'**, n = 4 controls, 4 mutants; **g–h'**, n = 5 controls, 5 mutants). (**i**) Quantification reveals statistically significantly reduced proliferation of *Wnt10a* mutant HF compared with controls. >20 control and >20 mutant HF were counted from n = 4 mutants and 4 controls (P18-29) or 5 mutants and 5 controls (P25-35). Significance was calculated with two-tailed Student's *t*-test. Error bars indicate s.e.m. (**j**) Accelerated catagen following *Wnt10a* deletion from P9 (photographed at P14 after hair clipping). (**k**) Delayed anagen entry following *Wnt10a* deletion from P20 (photographed at P26 after hair clipping). (**l,m**) Reduced *TL-GFP* expression in HFs (arrows) following *Wnt10a* deletion from P18. (**n,o**) *Wnt10a* mutant HFs are miniaturized and sebaceous glands (arrows) enlarged compared to controls at P180. (**p–s**) Oil-red O (**p,q**) and FABP5 (**r,s**) staining reveal increased lipid in *Wnt10a* mutant HFs compared with controls (arrows). (**t–w**) Bulge stem cell markers KRT15 and CD34 are retained in *Wnt10a* mutant HFs at P180. Scale bar, 25 µm (**l,m,r,s**), 50 µm (**a'–h',n–q,t–w**) or 200 µm (**a–h**).

filiform papillae and plantar epithelium (Fig. 6j–l and Supplementary Fig. 3v,w,z), similar to the effects of inducible β-catenin deletion[16]. Basal proliferation was also reduced in fungiform and circumvallate TBs (Fig. 6m–q).

Sweat gland ducts are composed of proliferating basal and non-proliferative suprabasal populations, both of which renew at least every 6 weeks[40]. Inducible *Wnt10a* or β-catenin deletion caused decreased sweat duct basal cell proliferation (Fig. 6x–z and

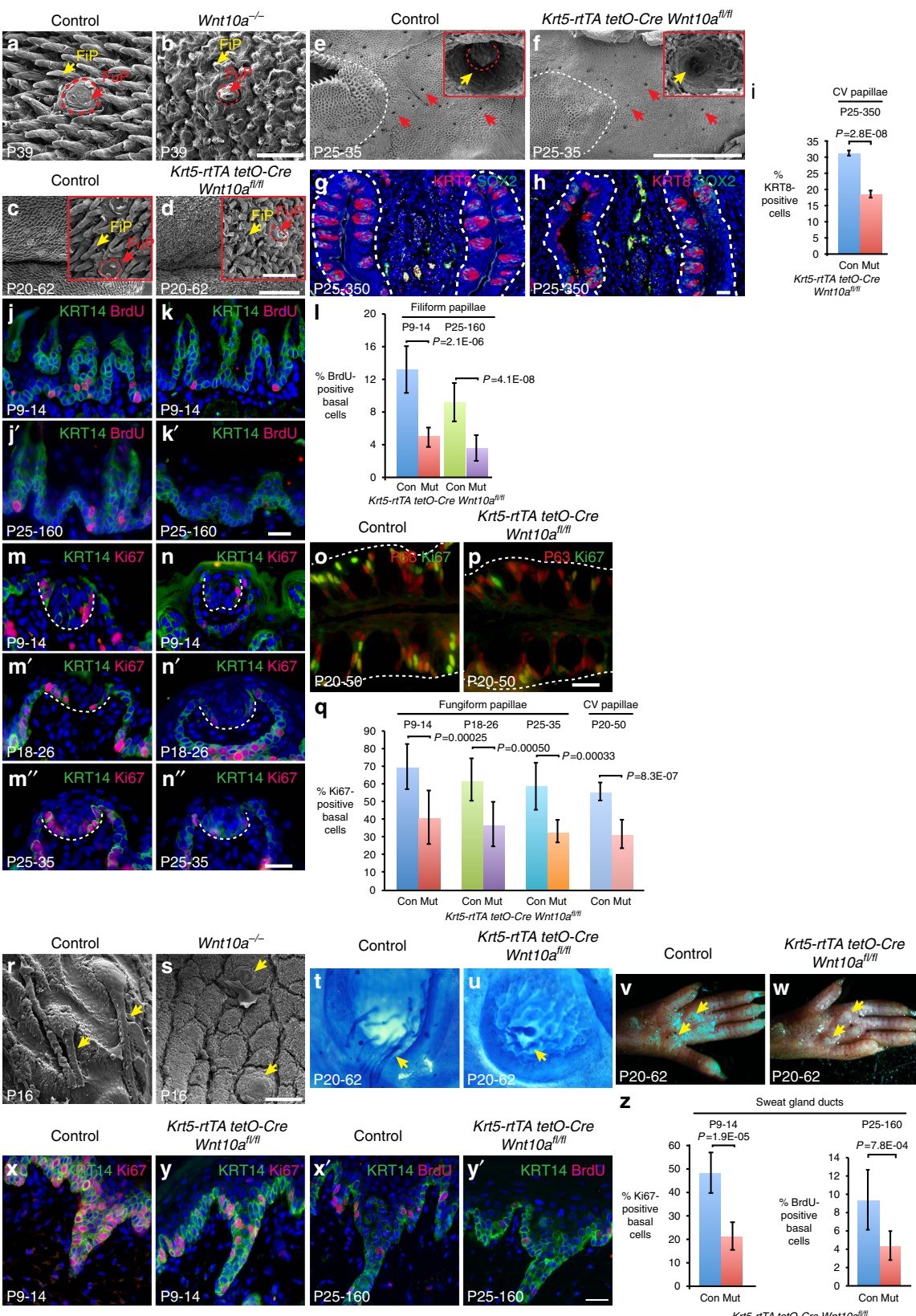

Supplementary Fig. 3x–z). Thus, WNT10A/β-catenin signalling is required for progenitor cell proliferation and adult renewal in non-hairy epithelia, as well as for normal hair growth.

**Axin2 marks self-renewing stem cells in adult epithelia.** Wnt-activated self-renewing stem cells are present in filiform papillae, interfollicular epidermis and resting (telogen) HFs[16,41,42]. However, whether Wnt activity marks self-renewing stem cells in anagen HFs is unknown. The Wnt/β-catenin target *Lgr5* marks stem cells in a subset of TBs in posterior tongue[43], but it is unclear whether Wnt activity is a universal characteristic of TB stem cells. The related gene *Lgr6* marks self-renewing stem cells for the sebaceous gland and nail[44,45], but it is unknown whether *Lrg6* is Wnt regulated at these sites, or whether Wnt-activated self-renewing cells are present in sweat gland ducts or dental epithelia. As epithelial progenitor cell proliferative defects contribute to *WNT10A* mutant phenotypes in these tissues we used lineage tracing to ask whether Wnt activity marks their stem cells.

To ensure that the time between labelling and long-term analysis was sufficient to ensure loss of initially labelled cells that were committed to terminal differentiation rather than self-renewal, we dox-induced *R26-rtTA tetO-H2B-GFP* mice from P14 to efficiently label histones in proliferating cells with H2B-GFP. Dox was withdrawn at P70 and tissues examined after 2 weeks or 3.5 months (P84 or P174). H2B-GFP label was retained in rarely cycling KRT15 + HF bulge stem cells at P84 and P174, but by P174 was absent from the SHG, upper follicle and sebaceous gland, and from sweat ducts and fungiform and circumvallate TBs (Fig. 7a–f). Thus, with the exception of the HF bulge, epithelial cells in all these tissues turned over within 104 days.

*Axin2^{lacZ}* and *TL-GFP* Wnt reporters are expressed in differentiated KRT8 + as well as basal TB cells. Lineage tracing in adult *Krt8-Cre^{ERT2} R26R^{EYFP} R26R-EYFP* mice tamoxifen treated for 2 days and examined after 4 days or 7 months revealed EYFP-labelled differentiated, but not basal, TB cells at 4 days, and complete loss of label by 7 months, indicating that KRT8 + cells are unable to self renew (Supplementary Fig. 5a,b).

*Axin2* is a ubiquitous Wnt/β-catenin target gene that provides a reliable indicator of Wnt/β-catenin pathway activity[42]. To determine whether Wnt signalling marks self-renewing stem cells, we utilized *Axin2-Cre^{ERT2/tdT}* mice[16] crossed with *R26R^{EYFP}*, *R26R^{mTmG}* or *R26R^{Confetti}* Cre reporter lines (Fig. 7g). Tissues were examined shortly after Cre induction to reveal cells active for Wnt signalling, and after several cycles of epithelial renewal to identify self-renewing stem cells and their

progeny. Un-induced *Axin2-Cre^{ERT2/tdT} R26R^{EYFP}* mice displayed sporadic EYFP labelling in palatal rugae and tongue muscle, but not in skin or tongue epithelia (Supplementary Fig. 5c). We did not detect leakiness in any of these tissues when *Axin2-Cre^{ERT2/tdT}* was used with *R26R^{mTmG}* or *R26R^{Confetti}*; however, as in any lineage tracing experiment, we cannot absolutely exclude that some clones resulted from leaky Cre activity.

In *Axin2-Cre^{ERT2/tdT} R26R^{mTmG}* mice induced in early anagen (P20-21) and examined 40 h after the first tamoxifen injection, mG + cells were detected in the HF SHG, KRT15 + bulge, DP and dermal sheath (Fig. 7h–j). After 15 days (P35, full anagen), mG + cells were present in both epithelial and dermal HF lineages (Fig. 7k–m). Cells in all regions of the permanent HF were labelled in telogen (Fig. 7n), and all lineages contained mG + cells in the subsequent anagen (Fig. 7o–q). Similar results were obtained using the *R26R^{Confetti}* reporter (Fig. 7r). Thus, self-renewing *Axin2*-expressing cells labelled in early anagen contribute to all epithelial and dermal HF components.

Sebaceous gland stem cells reside in the upper HF and sebaceous gland peripheral layer and are marked by LRIG1 and LGR6 (refs 44,46). After long-term lineage tracing with *Axin2-Cre^{ERT2/tdT} R26R^{mTmG}*, the isthmus and an entire lobe of the sebaceous gland was labelled in some HFs that lacked labelling of the bulge and lower HF (Fig. 7s). Examination of *Axin2-Cre^{ERT2/tdT} R26R^{Confetti}* lineage-traced skin, which permits clonal analysis, confirmed this result (Fig. 7t–v). Thus, *Axin2*-expressing cells in the isthmus and/or sebaceous gland peripheral layer repopulate the sebaceous gland during homoeostasis.

*Axin2-Cre^{ERT2/tdT} R26R^{EYFP}* fungiform and circumvallate papillae from mice induced at P20-P21 displayed EYFP expression in KRT14 + basal cells and KRT14 − differentiated TB cells 40 h after Cre induction (Fig. 8a,c,e,g,i). After 6 or 9 months, KRT14 + basal cells remained labelled, indicating that they self-renew (Fig. 8b,d), and we also detected EYFP in differentiated Type I and II cells in fungiform and circumvallate papillae and Type III cells in circumvallate papillae (Fig. 8b,d,f,h,j). Similar data were obtained using the *R26R^{mTmG}* reporter (Fig. 8k–n). Thus, self-renewing *Axin2*-expressing basal progenitors can produce all three taste cell types.

In *Axin2-Cre^{ERT2/tdT} R26R^{mTmG}* sweat gland ducts at 40 h after Cre induction, mG + cells were present in both basal (KRT6 − ) and suprabasal (KRT6 + ) layers (Fig. 8o), and both populations remained labelled after 4.5 months (Fig. 8p). As we did not detect significant proliferation of suprabasal cells, these likely originate from self-renewing *Axin2*-expressing basal cells.

Terminally differentiated cells of the nail arise from self-renewing KRT14 + LGR6 + stem cells in the proximal nail

**Figure 6 | WNT10A/β-catenin signalling is required for postnatal development and maintenance of epidermal appendages.** (**a**–**d**) SEM shows fungiform (FuP) and filiform (FiP) papilla defects in adult global (**a,b**) and inducible epithelial (**c,d**) *Wnt10a* mutants. (**e,f**) SEM reveals loss of TBs following inducible *Wnt10a* deletion in adult tongue epithelium. (**g**–**i**) Decreased expression of TB markers KRT8 (red) and SOX2 (green) (**g,h**) and reduced percentage of KTR8 + TB cells (**i**) in sectioned circumvallate papillae following inducible *Wnt10a* deletion. KRT8 + and total DAPI + cells were counted in 10 sections from 3 controls and 10 sections from 3 mutants. (**j**–**n''**,**o,p**) Induced *Wnt10a* deletion at the stages indicated causes decreased basal cell proliferation in filiform (**j**–**k'**), fungiform (**m**–**n''**) and circumvallate (**o,p**) papillae. (**l,q**) Quantification of proliferation in filiform (**l**), and fungiform and circumvallate (**q**) papillae. FiP: BrdU + /KRT14 + and total KRT14 + cells counted in 10 fields at 20 × from 3 controls and the same for 3 mutants. Fungiform TBs: Ki67 + /KRT14 + and total KRT14 + cells counted in 10 TBs from 3 control and 3 mutant (P9-14), 6 control and 6 mutant (P18-26) or 5 control and 5 mutant (P25-25) mice. Circumvallate TBs: Ki67 + /KRT14 + and total KRT14 + cells counted in 30 TBs from 3 controls and 30 TBs from 3 mutants. (**r,s**) SEM reveals failure of postnatal sweat duct development in P16 *Wnt10a^{−/−}* mutant footpad. (**t**–**w**) Inducible *Wnt10a* deletion in adults prevents sweat duct maintenance (Nile blue staining) (**t,u**) and decreases sweating (starch-iodine staining, purple dots) (**v,w**). (**x**–**y'**) Inducible *Wnt10a* deletion in early postnatal or adult life causes decreased sweat duct basal cell proliferation. (**z**) Quantification of sweat duct proliferation. Ki67 + /KRT14 + or BrdU + /KRT14 + and total KRT14 + cells counted in 10 ducts from 3 controls and 10 ducts from 3 mutants (P9-14) or 10 ducts from 4 controls and 10 ducts from 4 mutants (P25-160). ≥3 control and 3 mutants used for other analyses. Significance was calculated with two-tailed *t*-test. Error bars indicate s.e.m. Dox induction periods are indicated; mice were analysed at the end of the induction period. Scale bar, 25 μm (**j**–**p**,**x**–**y'**), 50 μm (**r,s**), 100 μm (**a,b,g,h**, insets in **c,d**), 500 μm (**c,d**) or 2 mm (**e,f**).

matrix[45,47]. Consistent with *Axin2^lacZ* expression in this region[47], we observed mG+ KRT14+ proximal matrix cells (Fig. 8q, yellow arrows) as well as mG+ differentiating cells (Fig. 8q, red arrow) 4 days after induction of *Axin2-Cre^ERT2/tdT R26R^mTmG* mice. Clones of mG+ cells emanating from the KRT14+ proximal matrix and giving rise to the nail bed and nail plate

persisted after 6 months (Fig. 8r, arrows). Thus, self-renewing *Axin2*-expressing basal cells contribute to nail growth.

Interestingly, we were unable to detect labelling of the epithelial incisor tooth cervical loop, the site of stem cells that generate enamel-secreting ameloblasts of the continuously growing incisor. This is consistent with absence of enamel defects in *Wnt10a*

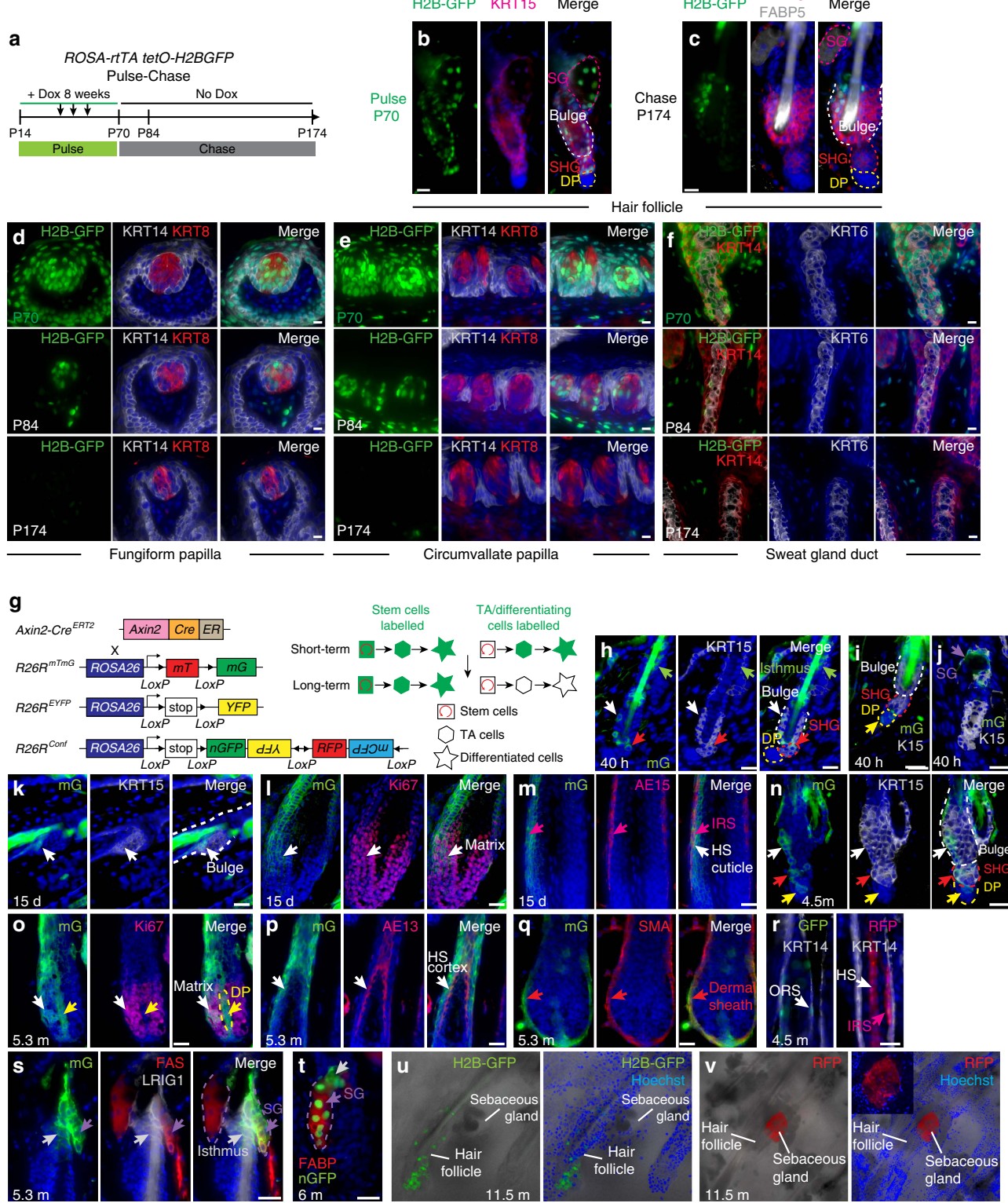

mutant incisors, and previous reports documenting lack of Wnt/β-catenin signalling in the cervical loop[48,49]. Thus, *Axin2* promoter activity is not a universal characteristic of self-renewing epithelial stem cells.

**WNT10A controls region-specific epithelial differentiation.** Patients and mice with *WNT10A* mutations exhibit severe tongue and palmoplantar abnormalities that are highly region specific, and cannot entirely be explained by decreased progenitor cell proliferation. We therefore asked whether WNT10A loss also affects specialized tissue differentiation. Remarkably, we found that expression of the hard keratin genes *Krt36* (*Krt1-5*) and *Krt84* (*Krt2-16*) that specifically characterize tongue filiform papillae was reduced or absent in *Wnt10a* mutant tongue, or following inducible epithelial β-catenin deletion, even though rudimentary papilla structures were present and included differentiating cells (Fig. 9a–d). This result is consistent with co-expression of *Axin2-tdT* with HOXC13, a transcription factor required for *Krt36* and *Krt84* expression[50] (Fig. 2i–i″). Nuclear β-catenin and TCF4 also localize to HOXC13 + cells, which are non-overlapping with LEF1+ proliferating progenitors (Supplementary Fig. 6a–d). Inducible epithelial *Wnt10a* or β-catenin deletion caused absence of nuclear, or all, β-catenin, respectively (Fig. 9e–h), and loss of LEF1 and HOXC13 expression (Fig. 9e′–m).

Human *WNT10A* patients display skin scaling and cracking of skin in palmoplantar, but not other, skin regions. These defects overlap with the effects of mutation or deletion of *KRT9*, whose expression is confined to suprabasal palmoplantar epidermis in humans[51] and footpad epidermis in mice[52]. *KRT9* is the most highly differentially expressed gene in human palmoplantar versus dorsal hand or foot epidermis, and *in vitro* studies suggest that Wnt/β-catenin signalling enhances its expression[53,54]. Consistent with this, we observed nuclear-localized β-catenin and TCF3 in mouse footpad epidermis (Supplementary Figs 6e,f and 7r). LEF1 expression was undetectable in this tissue, and TCF1 and TCF4 were mainly associated with sweat ducts (Supplementary Fig. 6g–i). Global *Wnt10a* loss or inducible epithelial β-catenin deletion caused decreased KRT9 protein and mRNA levels (Fig. 9n–r). By contrast, the suprabasal keratin gene *Krt10* and the terminal differentiation proteins filaggrin and involucrin, which locate to epidermis in all body regions, were unaffected by *Wnt10a* or β-catenin deletion (Supplementary Fig. 9j–r). Similarly, while KRT10 and loricrin expression was comparable in WNT10A patient and sex- and age-matched control plantar epidermis (Fig. 9s′–t″,u′–v″ and Supplementary Fig. 6s,t), KRT9 protein was decreased in our *WNT10A c.756 + 1 G > A* patient, and in an unrelated patient homozygous for a *WNT10A c.391 G > A*

mutation[4] (Fig. 9s–v″), and *KRT9* mRNA levels were lower in patient plantar skin than in control (Supplementary Fig. 6u).

**β-catenin complexes in differentiation and proliferation.** Our surprising finding that WNT10A/β-catenin signalling is required for region-specific differentiation as well as progenitor proliferation suggested that β-catenin may complex with transcriptional co-activators that are specifically expressed in differentiating versus proliferating cells, allowing its localization to distinct sets of target genes. We therefore examined potential transcriptional partners that are differentially expressed in suprabasal versus basal epithelial cells. A strong candidate was KLF4, which localizes to suprabasal epidermis, but is low or absent in the basal layer, and is required for embryonic tongue filiform papillae differentiation[7]. KLF4 expression overlapped with nuclear β-catenin and HOXC13 in adult tongue filiform papillae (Fig. 10a and Supplementary Fig. 7a), but was unaffected by loss of either *Wnt10a* (Fig. 10a) or β-catenin, indicating that it is not regulated by WNT10A/β-catenin signalling. Inducible epithelial *Klf4* deletion in adults caused defects in filiform papilla structure and loss of HOXC13, but did not affect LEF1 expression or basal proliferation (Supplementary Fig. 7b–e and Fig. 10b,c). Nuclear β-catenin remained detectable in LEF1-, as well as LEF1 + , filiform papilla cells in *Klf4* mutants (Fig. 10d); thus, nuclear β-catenin is insufficient for HOXC13 expression in the absence of KLF4. Consistent with this, mutation of epithelial β-catenin to a constitutively active form in *Krt5-rtTA tetO-Cre Ctnnb1^fl(Ex3)/ +* mice enhanced HOXC13 levels in differentiating filiform papilla cells, but did not induce its ectopic expression in basal cells (Supplementary Fig. 7f–i).

These data suggested that KLF4, β-catenin and TCF4 might co-regulate differentiation genes in filiform papillae. To test whether β-catenin, KLF4 and TCF4 form a complex in differentiating filiform papilla cell nuclei, we used proximity ligation assay (PLA), which detects interacting epitopes that are separated by less than 30–40 nm (ref. 55). PLA with antibodies to β-catenin and KLF4, TCF4 and KLF4 or β-catenin and TCF4 produced signals in differentiating cell nuclei but not in progenitors (Fig. 10e–g, yellow arrows). Conversely, PLA for β-catenin and LEF1 produced positive signals in proliferating basal cell nuclei (Fig. 10h, white arrows). Signals were not detected in tissue lacking β-catenin or *Klf4* (Fig. 10e–h and Supplementary Fig. 7j,k), or using only one antibody (Supplementary Fig. 7l,m), indicating specificity of the assay. β-catenin/LEF1 complexes formed in the absence of KLF4 (Supplementary Fig. 7n,o), and, interestingly, TCF4/KLF4 complex formation was unaffected by β-catenin deletion (Supplementary Fig. 7p,q).

To determine whether KLF4 plays a similar role in region-specific footpad epidermal differentiation, we examined its localization in this tissue. Immunofluorescence revealed KLF4

**Figure 7 | Pulse-chase estimates of epithelial tissue turnover rates and lineage tracing of Axin2 + cells in anagen HFs.** (**a–f**) Pulse-chase analysis of epithelial turnover times. (**a**) Labelling strategy: *R26R-rtTA tetO-H2B-GFP* mice were placed on doxycycline (dox) chow for 8 weeks (P14-P70) to induce expression of H2BGFP and its incorporation into the chromatin of dividing cells (Pulse). Mice were removed from dox treatment at P70 and analysed at successive time points following dox withdrawal (Chase). Label is gradually diluted out in cells that continue to proliferate. (**b,c**) Most HF epithelial cells, including sebaceous gland cells and some KRT15 + bulge stem cells, were H2BGFP positive at P70 (**b**). By P174, label had been lost from the epithelium, with the exception of KRT15 + bulge LRCs (**c**). (**d,e**) In fungiform (**d**) and circumvallate (**e**) taste papillae, epithelial cells including KRT14 + TB basal cells and KRT8 + differentiated TB cells were H2BGFP + at P70, but lost label by P174 indicating that they turned over within 104 days. (**f**) In sweat ducts, KRT14 + basal cells and KRT6 + luminal cells were H2BGFP + at P70, but had completely lost label by P174. (**g–v**) Lineage tracing of *Axin2*-expressing cells in HFs. (**g**) Schematic of lineage-tracing strategy. (**h–t**) Paraffin sectioned dorsal skin from *Axin^CreERT2/tdT^ R26R^mTmG^* (**h–q,s**) or *Axin^CreERT2/tdT^ R26R^Confetti^* (**r,t**) mice tamoxifen treated at P20-21 and analysed at the time points indicated. DAPI-stained sections were co-stained with markers for bulge and SHG (KRT15), proliferation (Ki67), inner root sheath (IRS) (AE15), hair shaft (HS) precursors (AE13), dermal sheath (SMA), outer root sheath (ORS) (KRT14), isthmus (LRIG1), sebaceous gland (SG) (FAS, FABP) as indicated. (**u,v**) Whole-mounted tail skin epidermis from *Axin^CreERT2/tdT^ R26R^Confetti^* mice induced at P28-29 and analysed at 11.5 months. At least three mice were analysed for each tracing condition; ≥20 labelled HFs were analysed for each mouse. Scale bar, 10 μm (**b–f**) or 25 μm (**h–t**).

expression in suprabasal footpad epidermis where it co-localized with TCF3 (Supplementary Fig. 7r,s), and was unaffected by β-catenin deletion (Fig. 10i). Inducible epithelial *Klf4* deletion in adults caused enhanced pigmentation and flaking of footpad skin (Supplementary Fig. 7t,u), a phenotype overlapping with the

effects of loss of KRT9 (ref. 56). In line with this, KRT9 protein and mRNA expression were significantly reduced in *Klf4* mutant footpad epidermis compared with controls, while *Krt10* expression was unaffected (Fig. 10j,k,q). As in dorsal tongue epithelium, nuclear and cytoplasmic β-catenin localization was

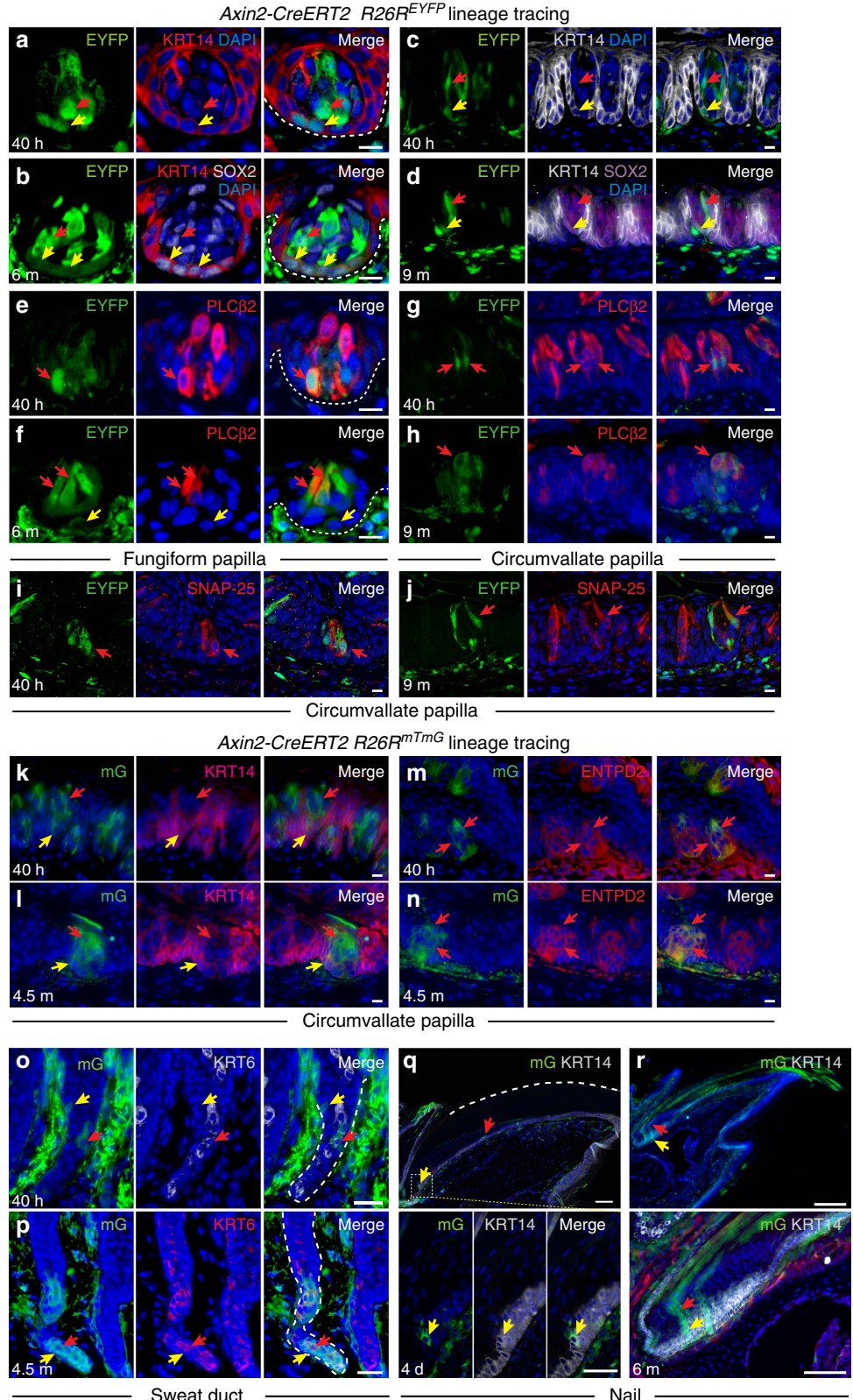

unaffected by *Klf4* deletion (Fig. 10l). PLA revealed direct interactions of TCF3 and KLF4, and β-catenin and KLF4, in suprabasal footpad epidermal nuclei (Fig. 10m,n), and direct interactions of β-catenin and TCF3 in both basal and suprabasal cells (Fig. 10o). Gain-of-function mutation of epithelial β-catenin enhanced β-catenin/TCF3 interactions (Fig. 10p). A consensus TCF/LEF-binding motif conserved in dog, horse and human localizes 16,185 bp downstream of the mouse *Krt9* transcription start site (TSS), close to a predicted KLF4-binding site. ChIP assays confirmed binding of KLF4, TCF3 and β-catenin to this region in footpad epidermis (Fig. 10r).

Our data support a model in which, in KLF4- basal cells, β-catenin promotes proliferation via interactions with LEF1 in filiform papillae and TCF3 in footpad epidermis. In suprabasal cells, which express KLF4 in a non-β-catenin-dependent manner, β-catenin instead complexes with KLF4 and TCF4 in filiform papillae, and with KLF4 and TCF3 in footpad epidermis, to activate specialized suprabasal differentiation programmes (Fig. 10s). The regional specificity of LEF1, TCF3 and TCF4 expression may underlie the distinct transcriptional programmes activated by WNT10A/β-catenin signalling in filiform papillae versus footpad epidermis.

## Discussion

Here we delineate the developmental and regenerative defects underlying ectodermal phenotypes in humans and mice carrying *WNT10A* mutations. We demonstrate that WNT10A/β-catenin signalling controls progenitor cell proliferation in a highly diverse set of epithelia characterized by distinct tissue organizations, differentiation pathways and cellular turnover rates. Using lineage tracing experiments with *Axin2-Cre^ERT2*, we identified self-renewing *Axin2*-expressing cells in the tissues affected by loss of WNT10A function. While we cannot exclude that additional pathways regulate *Axin2*, its expression in skin epithelia is removed by β-catenin deletion[42]. Thus, our data suggest that Wnt-active self-renewing stem cells contribute to regeneration of *Wnt10a*-dependent tissues. However, Wnt signalling is not universally associated with epithelial regeneration, as we were unable to detect *Axin2*-expressing progenitors in rodent incisor epithelium. Unexpectedly, in addition to directing normal embryonic tooth morphogenesis and adult progenitor cell proliferation, we find that WNT10A is required for region-specific differentiation, explaining the smooth tongue and palmoplantar keratoderma phenotypes observed in patients with *WNT10A* mutations.

Using genetic mouse models, we show that long-term absence of WNT10A causes HF miniaturization and sebaceous gland enlargement, with retention of bulge stem cells. This phenomenon is also observed in human androgenetic alopecia, consistent with identification of a *WNT10A* variant associated with lower expression levels and male pattern baldness. Functional studies of the *WNT10A* variant may shed further light on the aetiology of this common condition. Conversely, WNT10A's broad functions in epithelial proliferation suggest its possible requirement in skin squamous and basal cell tumours, which depend on β-catenin signalling[57,58]. In line with this, WNT10A overexpression *in vitro* promotes an invasive and self-renewing phenotype in oesophageal squamous cell carcinoma cells[59].

Sebaceous gland stem cells reside in the HF isthmus and sebaceous gland peripheral layer[44,46]. We find that these regions express several independent Wnt reporters, and contain *Axin2* + self-renewing stem cells that give rise to the sebaceous gland. Paradoxically, impaired Wnt/β-catenin signalling in *Wnt10a* mutants, and in mice expressing dominant negative LEF1 (ref. 31), causes sebaceous gland enlargement. Decreased signalling in the absence of *Wnt10a* may alter the balance between proliferation and differentiation of sebaceous gland stem cells. Alternatively, *Axin2* may simply mark sebaceous gland stem cells rather than reflecting a cell autonomous requirement for Wnt signalling, and sebaceous gland enlargement could be caused by altered bulge cell fate. Further experiments will be required to distinguish between these models.

Although *Wnt10a* expression localizes to embryonic appendage anlagen, we find that its requirements in embryogenesis are largely restricted to developing teeth, where it is required for normal levels of β-catenin signalling and *Shh* expression, explaining the tooth size and cusp abnormalities observed in mutant mice and human patients. In parallel, reduced expression of the direct Wnt target gene *Dlx3*, which is required for root formation, explains the root defects. The key roles of WNT10A in molar cusp and root development suggest potential utility of this molecule in dental engineering strategies.

While microdontia and taurodontism are concordant between human patients and mouse mutants, patients with *WNT10A* mutations also display secondary tooth agenesis. Lack of this phenotype in mouse mutants is not surprising, as mice do not develop a secondary dentition. Interestingly, some *Wnt10a* mutant mice develop ectopic molar M4 teeth that are loosely anchored and preferentially lost, a phenotype that has not been described in human WNT10A patients. Careful examination of the dentition in very young, severely affected patients might similarly reveal ectopic formation of primary molars. Alternatively, differences in the mechanisms controlling patterning of dental development in humans and mice could explain the absence of ectopic primary molar formation in human patients.

The mechanisms underlying repression of Wnt signalling in quiescent stem cells and its subsequent activation as cells enter a proliferative phase have been well studied[60]. However, how this pathway is controlled as proliferating cells make the critical transition to differentiation is much less clear. Using footpad and

---

**Figure 8 | Lineage tracing of Axin2 + cells in taste papillae and sweat ducts and nails.** (**a**–**j**) Adult *Axin2-Cre^ERT2/tdT* *R26R^EYFP* mice were tamoxifen treated at P60 and P61 and EYFP expression analysed in fungiform (**a**,**b**,**e**,**f**) and circumvallate (**c**,**d**,**g**–**j**) papillae after 40 h, 6 months or 9 months as indicated. KRT14 marks basal cells (**a**–**d**); SOX2 marks basal and Type I TB cells (**b**,**d**); PLCβ2 marks Type II TB cells (**e**–**h**); SNAP-25 marks Type III TB cells (**i**,**j**). Yellow arrows indicate EYFP + basal cells; red arrows indicate EYFP + differentiated TB cells. Both basal and differentiated TB cells remain labelled over 6–9 months. (**k**–**n**) Adult *Axin2-Cre^ERT2/tdT* *R26R^mTmG* mice were treated with tamoxifen on 2 successive days and mGFP expression analysed in cryosectioned circumvallate papillae at 40 h (**k**,**m**) and 4.5 months (**l**,**n**) as indicated. KRT14 marks basal cells (**k**,**l**); ENTP2 marks differentiated Type I TB cells (**m**,**n**). Both basal KRT14 + cells (yellow arrows) and differentiated KRT14 − cells, including ENTP2 + Type I cells, (red arrows) were labelled at 40 h and maintained label after 4.5 months. (**o**–**r**) Sectioned sweat ducts (**o**,**p**) and nail (**q**,**r**) from *Axin^CreERT2/tdT* *R26R^mTmG* mice tamoxifen treated at P20-21 and analysed at the stages indicated. Yellow arrows indicate KRT14 + basal cells; red arrows indicate KRT6 + sweat duct suprabasal cells (**o**,**p**) or KRT14 − differentiating nail plate cells (**q**,**r**). Insets in **q** are magnified images of boxed region; lower image in **r** is a higher magnification photomicrograph of the arrowed region in the upper image. At least three mice were analysed for each tracing condition; ≥10 labelled taste papillae or sweat gland ducts and ≥3 nails were analysed for each condition. Scale bar, 10 μm (**a**–**n**), 25 μm (**o**,**p**), 50 μm (insets in **q**), 100 μm (**q**; magnified region (lower image) in **r**) or 250 μm (upper image in **r**).

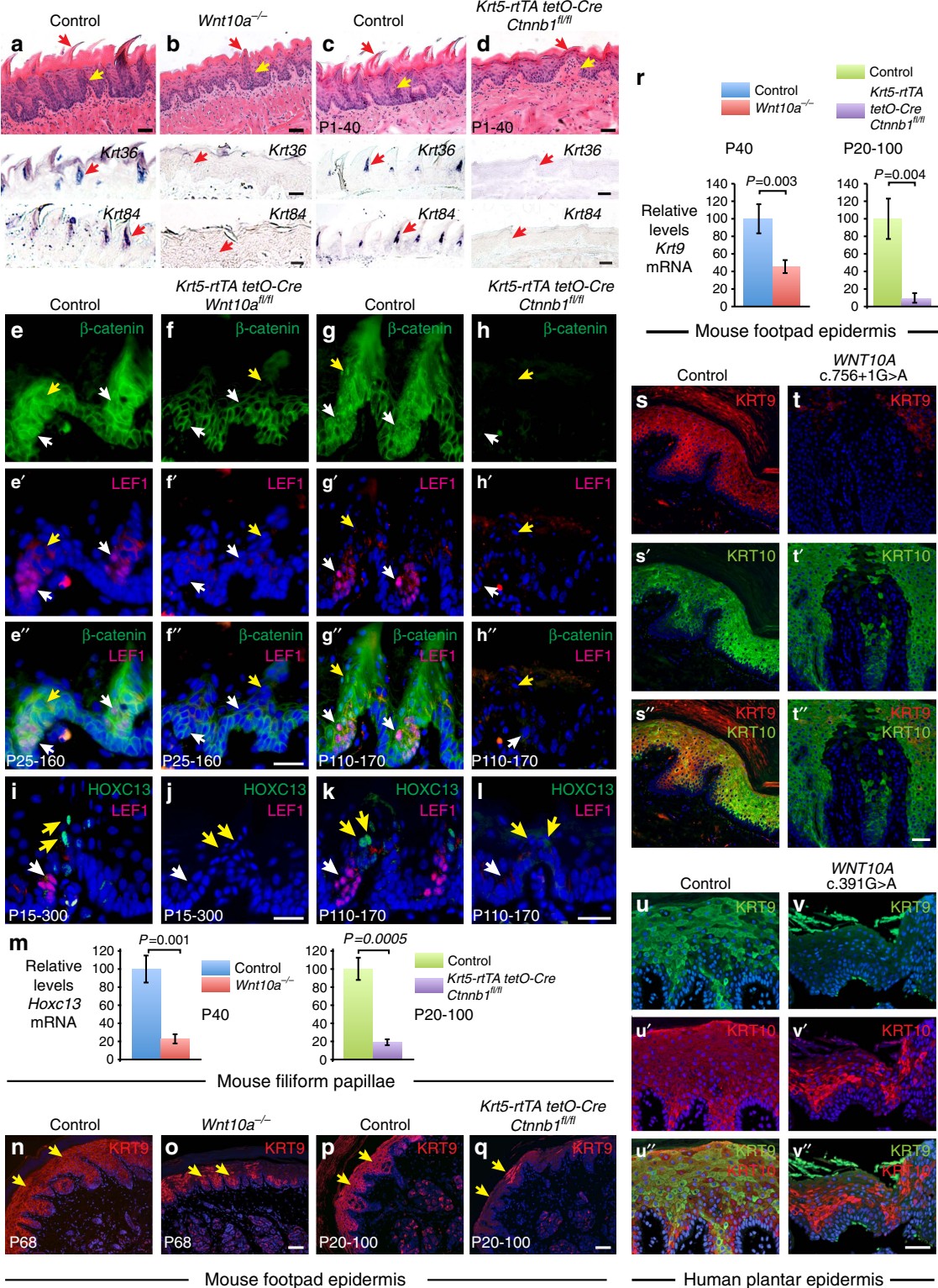

**Figure 9 | WNT10A/β-catenin signalling is required for region-specific differentiation.** (**a**–**d**) Filiform papillae are present in *Wnt10a⁻/⁻* and inducible β-catenin mutant dorsal tongue (yellow arrows), but horny structures and expression of hard keratins (*in situ* hybridization, purple signals) are decreased (red arrows). (**e**–**l**) Epithelial deletion of *Wnt10a* (**e**–**f″**,**i**,**j**) or β-catenin (**g**–**h″**,**k**,**l**) induced from P25, P110 or P15 as indicated causes decreased expression of nuclear β-catenin, LEF1 and HOXC13 (white arrows, LEF1+ proliferating cells; yellow arrows, HOXC13+ differentiating cells). (**m**) qPCR shows significantly decreased *Hoxc13* levels in *Wnt10a* and β-catenin mutant tongue epithelium. (**n**–**r**) IF and qPCR reveal reduced levels of KRT9 protein (**n**–**q**) and mRNA (**r**) in *Wnt10a⁻/⁻* and inducible β-catenin mutant footpad epidermis. (**s**–**v″**) Co-IF for KRT9 and KRT10 in plantar epidermis from patients homozygous for *WNT10A c.756+1G>A* (**s**–**t″**) or *WNT10A c.391G>A* (**u**–**v″**) compared with similarly aged sex-matched controls. For qPCR, RNA levels were quantified in six control and six mutant (P40) or four control and four mutant (P20-100) samples with three technical replicates for each, and normalized to β-actin mRNA. Significance was calculated with two-tailed Student's *t*-test. Error bars indicate s.e.m. Scale bar, 25 μm (**e**–**l**) or 50 μm (**a**–**d**,**n**–**q**,**s**–**v″**).

tongue epithelia as relatively simple model systems, we find that inducible epithelial *Klf4* deletion mimics region-specific differentiation, but not proliferative, defects seen in the absence of *Wnt10a* or epithelial β-catenin. Our data indicate that in basal cells, where KLF4 is low or absent, nuclear β-catenin complexes with LEF1 in filiform papillae and with TCF3 in footpad epidermis to control basal proliferation. By contrast, in

differentiating cells, nuclear β-catenin directly associates with KLF4 and TCF4 in filiform papillae, and with TCF3 and KLF4 in footpad epidermis to regulate expression of region-specific differentiation genes. This mechanism may be generalizable to other specialized epithelia, such as HFs, TBs, sweat ducts and nails, that exhibit β-catenin signalling in differentiating as well as proliferating cells. In line with this, in addition to controlling HF

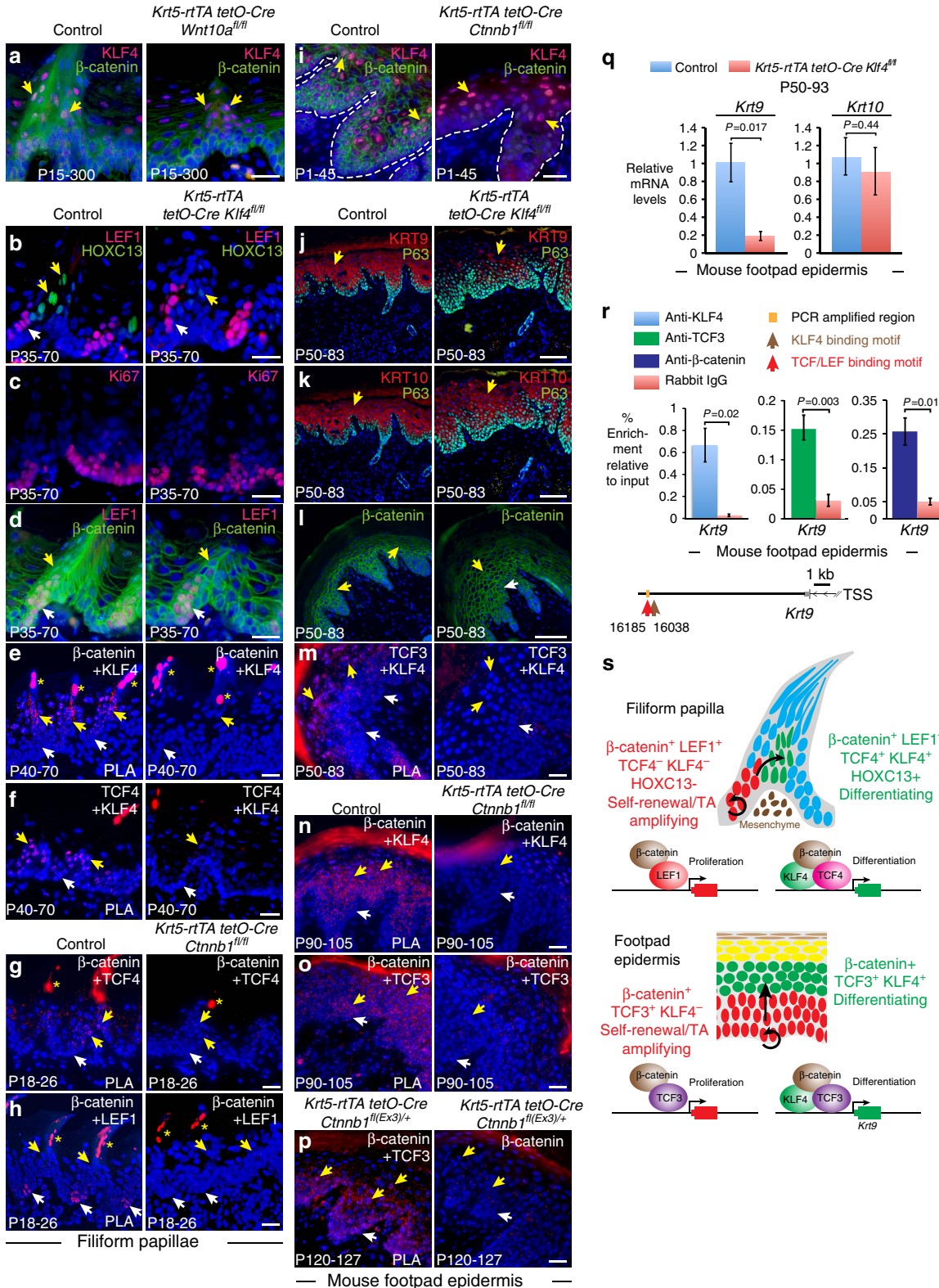

progenitor cell proliferation[16], Wnt/β-catenin signalling has been implicated in the regulation of hair keratin gene expression[61,62], and we observed disorganized internal hair shaft structures in *Wnt10a* mutant mice.

While KLF4 is broadly expressed in suprabasal cells and plays general roles in epidermal differentiation[8], we identified region-specific differences in the expression of LEF/TCF family members in filiform papillae versus footpad epidermis. These may contribute to the ability of β-catenin/KLF4/LEF/TCF complexes to direct region-specific differentiation programmes. Global analysis of the target genes activated by these distinct complexes will be a fascinating area for further study.

Our finding that KRT9 expression is regulated by WNT10A through the β-catenin pathway has important therapeutic implications for *WNT10A* patients. Aside from dental defects that require corrective surgery, palmoplantar fissuring is the most troubling clinical manifestation in these patients and results in significant discomfort. Given the known role of KRT9 in ensuring integrity of palmoplantar epidermal cells, it is likely that decreased KRT9 is an important contributor to this phenotype. Current treatments, including topical urea to soften the epidermis and steroids to reduce inflammation secondary to epidermal fragility, are non-specific and poorly effective[4]. Our data suggest topical application of appropriately formulated small molecule Wnt/β-catenin pathway activators as a potentially more effective approach to treat this condition by enhancing KRT9 levels in palmoplantar epidermis.

## Methods

**Human tissues.** *WNT10A* patients provided informed consent for publication of photographs, taste testing and collection of skin samples. All analyses of human patients were carried out under protocols approved by the IRB of the University of Pennsylvania, Philadelphia, PA 191904, USA and St Thomas' Hospital Ethics Committee, Westminster Bridge Rd, London SE1 7EH, UK. Normal controls were exempt de-identified discarded tissue provided by the University of Pennsylvania Skin Disease Research Center with IRB approval. Specimens were formalin-fixed, embedded in paraffin and sectioned at 5 μm.

**Starch-iodine sweat testing.** Iodine/alcohol (2 g iodine per 100 ml ethanol) solution was applied to human palm or to the plantar surface of a mouse rear paw with a brush. Once dry, the skin surface was painted with starch-oil suspension (100 g starch per 100 ml castor oil) and imaged after 2–3 min (mouse) or 20 min (human).

**Human taste testing and analysis of bitter receptor genes.** For taste test outcome measures, the patient's ratings were compared with data from 41 women unselected for health complaints and born within 5 years of the patient. In the taste test, subjects rated six taste solutions twice for intensity (water, sucrose, sodium chloride, quinine, phenylthiocarbamide (PTC) and denatonium benzoate)[63]. Each of the two ratings is referred to as a trial, that is, trial 1 and trial 2. Subjects also

indicated the taste quality descriptor, which best reflected the tasted solution, that is, water has little or no discernable flavour, sucrose is sweet, sodium chloride is salty and the other three compounds tested are perceived as bitter by most people. Genomic DNA was extracted from a saliva sample for genotyping for the bitter receptor gene for quinine sensitivity, *TAS2R19* and PTC sensitivity receptor *TAS2R38i* using TaqMan SNP genotyping assay kits (Thermo Fisher Scientific, Waltham, MA).

**Quantitative RT–PCR.** For human samples, total RNA was extracted from paraffin sections using proteinase K digestion, phenol–chloroform extraction and ethanol precipitation. For mouse tissues, total RNA was extracted using the RNeasy plus kit (Qiagen, Valencia, CA). RNA was treated with Turbo DNA-free Kit (Thermo Fisher Scientific) to remove genomic DNA. Treated RNA was reverse-transcribed using SuperScript III First-Strand kit (Thermo Fisher Scientific). qPCR was performed using a StepOnePlus Real-Time PCR System with Power SYBR Green master mix (Thermo Fisher Scientific). Relative expression levels were standardized against β-actin, *Gapdh* or *Krt10* as an internal control. At least three biological replicates were assayed in all experiments with mouse tissues. Three technical replicates were performed for each sample. Statistical significance was calculated using two-tailed Student's *t*-test. For experiments with human patient and normal control samples, three technical replicates were performed, but statistical significance could not be calculated as only one human patient sample was available for RNA extraction. Primers are listed in Supplementary Table 1.

**Experiments utilizing genetically modified mice.** Mice were maintained on a mixed strain background and were genotyped by PCR of tail biopsy DNA. Analyses of the HF growth cycle were performed in male mice due to the variability of the growth cycle in female mice. Both male and female mice were used for other analyses, and no obvious sex differences were noted. Animals were not excluded from analyses, except in the case of brief-access taste-testing experiments (see below). Experiments were performed unblinded with the exception of quantification of proliferation, and analysis of PLAs, which were carried out by researchers who were not informed of the sample genotypes. Randomization was not used as mice were assigned to control or experimental groups according to their genotype. We utilized $n = 5$ control and $n = 5$ experimental mice wherever possible, providing 80% power at a two-sided significance level of 0.05 to detect a difference (effect size) of 2.0 s, where s is the s.d.; at a minimum we utilized $n = 3$ control and $n = 3$ experimental mice, providing 80% power at a two-sided significance level of 0.05 to detect a difference of 2.8 s. All experiments were performed under animal protocols approved by the University of Pennsylvania IACUC Committee according to University of Pennsylvania guidelines.

**Mouse lines.** *Krt5-rtTA* mice[16] were provided by Dr Adam Glick, Pennsylvania State University. *Krt14-Cre* mice[64], *tetO-Dkk1* mice[16] and *Krt14-Krm1* mice[16] were generated in the Millar laboratory. *Axin2-Cre^{ERT2/tdT}* mice[16] were generated in the Morrisey laboratory. *Ctnnb1^{(Ex3)fl/+}* mice[65] were provided by Dr Makoto Taketo, Kyoto University. *Klf4^{fl/fl}* mice[66] were generated in the Kaestner laboratory. *tetO-Cre*, *Ctnnb1^{fl/fl}*, *Axin2^{LacZ}*, *R26-M2rtTA (R26-rtTA), CMV-Cre, TCF/LEF:H2B-GFP (TL-GFP), Gt(ROSA)26Sor^{tm4(ACTB-tdTomato,-EGFP)Luo}/J (R26R^{mTmG}), Gt(ROSA)26Sor^{tm1(CAG-Brainbow2.1)Cle}/J (R26R^{Confetti}), Gt(ROSA)26Sor^{tm1(EYFP)Cos}/J (R26R^{EYFP}), tetO-H2B-GFP* and *Krt8-Cre^{ERT2}* mice were obtained from Jackson Laboratories (Bar Harbor, ME). *Wnt10a^{fl/fl}* mice were generated using a modified recombineering procedure[67] to construct a gene-targeting vector with *loxP* sites flanking exons 3 and 4. 129X1/SvJ ES cell clones were screened by long-range PCR, confirmed by Southern blotting, and injected into mouse blastocysts to generate

**Figure 10 | β-catenin interacts differentially with TCF/LEF1 and KLF4 to control proliferation and specialized differentiation.** (**a**) Nuclear β-catenin and KLF4 overlap in differentiating filiform papillae cells (yellow arrows). Nuclear β-catenin, but not KLF4, is depleted following inducible *Wnt10a* deletion. (**b–d**) Inducible epithelial *Klf4* deletion causes decreased HOXC13 (**b**, green) but does not affect LEF1 (**b,d**, pink), proliferation (Ki67) (**c**, pink) or nuclear β-catenin (**d**, green). (**e–h**) PLA (pink signals) with adult dorsal tongue sections and anti-β-catenin and KLF4 (**e**), TCF4 and KLF4 (**f**), β-catenin and TCF4 (**g**), or β-catenin and LEF1 (**h**) reveals direct interactions of β-catenin, KLF4 and TCF4 in differentiating (yellow arrows) and β-catenin and LEF1 in proliferating (white arrows) cells. Signals were lacking following inducible *Klf4* or β-catenin deletion (right panels). Asterisks indicate autofluorescence. (**i**) Overlap of nuclear β-catenin and KLF4 in suprabasal footpad epidermis. KLF4 is unaffected by inducible β-catenin deletion. (**j–l**) Inducible epithelial *Klf4* deletion causes decreased expression of KRT9 (**j**) but not KRT10 (**k**) or nuclear β-catenin (**l**). (**m–p**) PLA in adult footpad with anti-TCF3 and KLF4 (**m**), β-catenin and KLF4 (**n**) or β-catenin and TCF3 (**o,p**) shows KLF4, β-catenin and TCF3 directly interact in suprabasal cells (yellow arrows); β-catenin and TCF3 interact in basal cells (white arrows). Signals are absent in tissues with inducible *Klf4* (**m**) or β-catenin (**n,o**) deletion (right panels), and are enhanced by inducible stabilizing mutation of β-catenin (**p**; right panel shows single antibody control). (**q**) qPCR shows significant reduction of *Krt9* but not *Krt10* mRNA levels in P93 footpad epidermis following inducible epithelial *Klf4* deletion. mRNA levels normalized to β-actin. (**r**) ChIP-qPCR with adult footpad epidermis shows specific enrichment for KLF4, TCF3 and β-catenin in a region containing conserved KLF4 and LEF/TCF-binding motifs, 16 kb downstream of *Krt9* TSS. Samples from ≥ 3 mutant and ≥ 3 control mice analysed in all assays. For **q,r**, three technical replicates were additionally performed for each sample; significance calculated with two-tailed Student's *t*-test; error bars indicate s.e.m. (**s**) Model showing distinct nuclear β-catenin complexes regulating proliferation versus differentiation in tongue and footpad epithelia. Scale bar, 25 μm (**a–i,m–p**) or 75 μm (**j–l**).

chimeric mice. *Wnt10a^fl/fl* mice were crossed with *CMV-Cre* mice (Jackson Laboratories) to generate global null mutants, and to *Krt5-rtTA tetO-Cre* or *Krt14-Cre* mice to obtain inducible *Krt5-rtTA tetO-Cre Wnt10a^fl/fl* or constitutive epidermal *Krt14-Cre Wnt10a^fl/fl* mutants. *Krt5-rtTA tetO-Cre Wnt10a^fl/fl* and *Wnt10a^−/−* mice were crossed with *TCF/LEF:H2B-GFP* (TL-GFP, Jackson Laboratories) mice to monitor Wnt pathway activity. For dox-inducible *Klf4*, *Wnt10a* or β-catenin mutation, or forced expression of *H2B-GFP* or *Dkk1*, mice were placed on $6 \text{ g kg}^{-1}$ dox chow (Bio-Serv, Flemington, NJ). For lineage tracing experiments, *Axin2-Cre^ERT2/tdT* mice were crossed with *Gt(ROSA)26Sor^tm4(ACTB-tdTomato,-EGFP)Luo)/J*, *Gt(ROSA)26Sor^tm1(CAG-Brainbow2.1)Cle/J* or *Gt(ROSA)26Sor^tm1(EYFP)Cos/J* Cre reporter alleles (Jackson Laboratories). *Krt8-Cre^ERT2* mice (Jackson Laboratories) were crossed with *Gt(ROSA)26Sor^tm1(EYFP)Cos* Cre reporter mice. Tamoxifen was administered by intraperitoneal injection at $0.2 \text{ mg g}^{-1}$ per day for 1–3 days to induce Cre activity.

**Micro-CT analysis.** Micro-CT imaging was performed using an eXplore Locus SP micro-CT specimen scanner (GE Healthcare, London, Ontario). Mouse jaws were fixed in 4% paraformaldehyde (PFA), wrapped in cloth gauze and immersed in PBS for scanning. Scan parameters were as follows: 80 kVp per 80 µA X-ray tube voltage/current, 250 µm Al filter, 400 views, 0.5° increment, 1.7 s exposure, 4 averages and 1 h scan time. Images were reconstructed at a resolution of 16 µm isotropic voxels on the same intensity scale (Hounsfield units) to allow comparison of relative apparent bone density. Colour volume renderings and multi-planar reformatted slices were generated using OsiriX (www.osirix-viewer.com) software, and maximum intensity projections were performed using FIJI (ImageJ, NIH) software.

**Histology and immunofluorescence.** Paraffin sectioned and cryosectioned tissue was subjected to haematoxylin and eosin staining or immunofluorescence. For immunofluorescence assays, tissues were fixed in 4% PFA, paraffin embedded and sectioned at 5 µm, or embedded in optimal cutting temperature embedding medium and cryosectioned at 10 µm. Paraffin sections were de-waxed through graded xylene solutions, microwaved in 10 mM sodium citrate and incubated with primary antibodies, followed by incubation with Alexa Fluor-labelled secondary antibodies (Thermo Fisher Scientific)[16]. For BrdU labelling, mice were injected with 50 µg BrdU $\text{g}^{-1}$ body weight and killed after 1–2 h. Skin samples were fixed in 4% PFA, embedded in paraffin, de-waxed and incubated with anti-BrdU antibody (Rockland Immunochemicals, Limerick, PA, 600-401-C29, 1:100)[16]. For tail skin whole mounts, skin samples were incubated with Dispase II and separated epidermis was counterstained with Hoechst 33342 (Thermo Fisher Scientific) and imaged on a Leica TCS SP8 confocal microscope (Leica Microsystems, Buffalo Grove, IL). For analyses of proliferation in slides stained for Ki67 or BrdU, sections were counterstained with 4,6-diamidino-2-phenylindole (DAPI). Sectioned tongue and footpad tissues were subjected to co-immunofluorescence for KRT14 to identify basal cells.

**Quantification and statistical analyses of immunostaining data.** For quantification of KRT8-positive cells in circumvallate papillae, total DAPI-stained nuclei and KRT8-positive cells were counted in 10 sections of circumvallate papillae from $n = 3$ mutant and 10 sections of circumvallate papillae from $n = 3$ control mice, and the percentage of KRT8-positive cells per papilla was calculated. For quantification of proliferation in HFs, total DAPI-stained nuclei and BrdU-positive nuclei were counted in at least 20 control and 20 mutant HF bulbs from $n = 4$ mutants and 4 controls (P18-29) or 5 mutants and 5 controls (P25-35). For filiform papillae and plantar epidermis, BrdU or Ki67-positive KRT14+ cells and total KRT14+ nuclei were counted in 10 fields (filiform papillae) or 15 fields (plantar epidermis) at × 20 magnification from three control mice and the same for three mutant mice. For fungiform TBs, Ki67-positive KRT14+ cells and total KRT14+ nuclei were counted in three control and three mutant (P9-14), six control and six mutant (P18-26) or five control and five mutant (P25-25) mice; a total of 10 TBs was counted for control and 10 for mutant at each stage. For circumvallate TBs, Ki67-positive KRT14+ cells and total KRT14+ nuclei were counted in 30 TBs from 3 control mice and 30 TBs from 3 mutant mice. For sweat ducts, Ki67 or BrdU-positive KRT14+ cells and total KRT14+ cells were counted in 10 ducts from 3 control mice and 10 ducts from 3 mutant mice at each stage (P9-14, P100-P160), or in 10 ducts from 4 control and 10 ducts from 4 mutant mice (P25-160). Significance was calculated with two-tailed *t*-test. For lineage tracing at least three mice were analysed for each tracing condition; ≥20 labelled HFs were analysed for each mouse; ≥10 labelled taste papillae or sweat gland ducts and ≥3 nails were analysed for each condition.

**Antibodies.** The following antibodies were used: mouse anti-β-catenin (Sigma-Aldrich Co., St. Louis, MO, C-7207, 1:1,000); rabbit anti-β-catenin (Novus Biologicals, Littleton, CO, NBP1-32239, 1:200); goat anti-β-catenin (R&D Systems, Minneapolis, MN, AF-1329, 1:100); rabbit anti-KLF4 (Cell Signaling Technology, Danvers, MA, 4038 S, 1:100); goat anti-KLF4 (R&D Systems, AF-3158, 1:50); rabbit anti-KRT 8 (Sigma-Aldrich Co., sab4501654, 1:200); mouse anti-KRT8 (Santa Cruz Biotechnology, Dallas, TX, sc-101459, 1:100); mouse anti-p63 (Santa Cruz Biotechnology, sc-8431, 1:100); rabbit anti-p63 (Cell Signaling Technology, 4892, 1:100); mouse anti-KRT14 (Abcam, Cambridge, UK, Ab7800-500, 1:200); rabbit anti-KRT14 (Covance, Princeton, NJ, 905301, 1:1,000); rabbit anti-KRT10 (Covance, PRB-159 P, 1:1,000); mouse anti-KRT15 (Vector Laboratories, Burlingame, CA, VP-C411, 1:100);

rabbit anti-FAS (Abcam, ab22759, 1:200); goat anti-LRIG1 (Novus Biologicals, AF3688-SP, 1:100); mouse anti-hair cortex keratin (AE13) (Abcam, ab16113, 1:200); rabbit anti-KRT6 (Covance, PRB-169 P, 1:1,000); rabbit anti-COL17A1 (Abcam, Ab184996, 1:200); rabbit anti-filaggrin (Covance, PRB-417 P, 1:1,000); rabbit anti-loricrin (Covance, PRB-145 P, 1:1,000); rabbit anti-involucrin (Covance, PRB-140C, 1:200); rabbit anti-ENTPD2 (mN2-36 L, Centre Hospitalier Universitaire de Québec; 1:200); rabbit anti-PLCβ2 (Santa Cruz Biotechnology, sc-206, 1:200); rabbit anti-SNAP-25 (Sigma-Aldrich Co., S9684, 1:500); rabbit anti-Ki67 (Abcam, ab16667, 1:200); rabbit anti-LEF1 (Cell Signaling, 2286, 1:100); mouse anti-SMA (Sigma-Aldrich Co., a2547, 1:5,000); chicken anti-GFP (Cell Signaling, ab13970, 1:1,000); goat anti-FABP5 (R&D systems, BAF1476, 1:200); rabbit anti-BrdU (Rockland Immunochemicals, 600-401-C29, 1:100); mouse anti-HOXC13 (Abnova, Taipei, Taiwan, H00003229-M01, 1:100); rabbit anti-DLX3 (gift from Maria I. Morasso, 1:200); rabbit anti-mouse KRT9 (gift from Deena M Leslie Pedrioli, 1:200); mouse anti-human KRT9 (Thermo Fisher Scientific, MA1-35569, 1:10); rabbit anti-Cyclin D1 (Cell Signaling Technology, ab16663, 1:100); rabbit anti-SOX2 (Cell Signaling Technology, 4900, 1:100); rabbit anti-RFP (Rockland Immunochemicals, 600-401-379, 1:500); rat anti-CD34 (BD Biosciences, San Jose, CA, 553731, 1:40); rabbit anti-TCF3 (EMD Millipore, Billerica, MA, ABE414, 1:800); rabbit anti-TCF1 (Cell Signaling Technology, 2203, 1:100); and rabbit anti-TCF4 (Cell Signaling Technology, 2565, 1:100).

**RNA *in situ* hybridization.** For RNA *in situ* hybridization with tissue sections, tissues were fixed in 0.4% PFA overnight, embedded in paraffin or cryoprotected and embedded in optimal cutting temperature embedding medium and sectioned. *In situ* hybridization assays were carried out using antisense digoxigenin-labelled RNA probes for β-catenin, *Shh*, *Wnt10a*, *Krt1-5*, *Krt2-16* and *Pax9* (refs 16,29,68). Probe templates for β-catenin (nt 150–540, NM_007614), *Wnt10a* (nt 1295–2487, U61969), *Krt1-5* (nt 374–724, X65506), *Krt2-16* (nt 520–745, AY028607) and *Pax9* (nt 1431–1643, NM_011041) were synthesized by PCR of embryonic mouse cDNA, with primers containing T7 RNA polymerase-binding sites. *Shh* hybridization probe was synthesized from a cDNA-containing plasmid[23]. After overnight hybridization, the slides were washed with saline sodium citrate buffer and maleic acid buffer and incubated with anti-digoxigenin alkaline phosphatase at 4 °C overnight. The signals were developed with BCIP-NBT or BM purple system (Roche, Basel, Switzerland). For whole-mount *in situ* hybridization, embryos were fixed in 4% PFA overnight and incubated with 1 µg digoxigenin-labelled riboprobes in 1 ml hybridization buffer at 70 °C. After overnight incubation, embryos were washed in saline sodium citrate and maleic acid buffers and incubated with anti-digoxigenin alkaline phosphatase at 4 °C overnight. After 48 h of washing with 0.1% Triton in PBS (PBST), the signals were detected by immersing embryos in BM purple solution (Roche).

**Whole-mount dye exclusion assay and immunofluorescence.** For dye exclusion assays, collected tissues were placed in PBS containing the following methanol concentrations for 2 min each: 0, 25, 50, 75, 100, 75, 50, 25 and 0%. Fixed tissues were immersed in 0.1% toluidine blue solution for 2 min, and washed in PBS[69]. For whole-mount immunofluorescence, mouse tongues were dissected and rinsed briefly in PBS. Excess muscle and ventral tissues were removed using a razor blade. The remaining dorsal tissues were flattened on a paper towel and placed in Dispase II ($2 \text{ mg ml}^{-1}$ in PBS) overnight at 4 °C. The following day, epidermal tissues were separated, washed briefly in PBS, fixed in 4% PFA for 60 min at 4 °C, washed in PBS, permeabilized in 1% Triton in PBS for 60 min, blocked in 1% Triton, 10% goat serum and Mouse On Mouse blocking reagent (Vector Laboratories, Burlingame, CA) in PBS for 2 h, and then incubated with primary antibody overnight at 4 °C. Samples were washed for 3 h with PBST, incubated with secondary antibody overnight at 4 °C, washed with PBST, mounted and imaged.

**Oil Red O staining.** For Oil Red O staining, cryosections were cut at 10 µm, fixed in 4% PFA, rinsed with 60% isopropanol and stained with Oil Red O working solution using an Oil Red O Stain Kit (American MasterTech, Lodi, CA) according to the manufacturer's instructions. Slides were counterstained with haematoxylin.

**SEM analysis.** Samples were fixed overnight in 4% PFA, dehydrated in graded ethanols, incubated for 20 min in 50% hexamethyldisilazane (Sigma-Aldrich Co.) in ethanol followed by three changes of 100% hexamethyldisilazane, air-dried overnight, mounted on stubs and sputter coated with gold palladium. Specimens were observed and photographed using a Quanta 250 scanning electron microscope (FEI, Hillsboro, OR, USA) at 10 kV accelerating voltage.

**Whole-mount Nile blue staining.** Dissected mouse ventral paw skin was placed in $2 \text{ mg ml}^{-1}$ Dispase II (Thermo Fisher Scientific) overnight at 4 °C. Epidermis was dissected with forceps and stained in 0.1% Nile blue solution for 2 min. Samples were washed with PBS, fixed in 4% PFA and imaged on a Leica MZ16F dissection microscope (Leica Microsystems, Buffalo Grove, IL).

**Two-bottle preference tests and brief-access gustometer tests.** We used two-bottle choice tests and brief-exposure tests, respectively, to assess preference and acceptance of taste solutions. For the two-bottle preference tests, cohorts of 8–14 $Wnt10a^{+/+}$ and 9–16 $Wnt10a^{-/-}$ littermates received ascending concentration series for 48 h each of two sweet compounds (sucrose and saccharin), two bitter compounds (quinine hydrochloride (QHCl) and denatonium benzoate), two sour compounds (HCl and citric acid), NaCl and capsaicin.

We carried out brief-access tests on several groups of 7–14 $Wnt10a^{+/+}$ and 7–16 $Wnt10a^{-/-}$ littermates. Mice were weighed daily for the duration of the experiment, immediately before their placement in an MS160-Mouse gustometer (DiLog Instruments, Inc., Tallahassee, FL). Each gustometer included a $14.5 \times 30 \times 15$ cm test chamber with a motorized shutter to control taste solution access. Taste solutions were contained in bottles on a rack that was mechanically controlled to provide the mouse with access to any one of eight specific taste solutions. For each bottle, the drinking spout was connected with a high-frequency alternating current contact circuit. This permitted detection and recording of each lick of the spout[70]. Zero concentration and washout trials (see below) utilized deionized water. Reagent grade taste compounds (Sigma-Aldrich Co.) were dissolved in deionized water at room temperature.

To control for any instrument-specific variation, repeat tests on each mouse were always conducted with the same gustometer and equal numbers of wild-type and mutant mice were tested with each of our eight gustometers. Mice were trained to sample taste solutions during 5 s access periods by depriving them of water for 22.5 h, placing them for 25 min in a gustometer with the shutter open, returning them to their home cage and supplying them with water ad libitum for 1 h. This cycle was repeated for 2 more days, with the exception that in the two subsequent cycles, the access shutter was closed 5 s after each time the mouse started drinking and was then re-opened after 7.5 s. All tested mice were successfully entrained using these procedures.

We administered each of the eight compounds to the control and mutant mice in three test sessions over two cycles. Responses to sucrose, quinine hydrochloride, HCl and NaCl were assayed in the first cycle, and the remaining compounds were tested in the second cycle. One 25 min session was conducted per day, and tested only one compound. For the hedonically positive compounds sucrose and saccharin[70], mice received free access to food and water for 24 h followed by 2 ml of water and 1 g of food, and were tested 24 h later. Mice then received unlimited access to food and water for 24 h before being prepared for the next test. Following testing with distasteful (hedonically negative) compounds, mice received unlimited water for 1 h before being deprived of water to prepare for the next session. For testing with hedonically positive compounds, the highest concentration was administered first to interest the mouse in accessing the drinking fluid. We then presented, in random order, once each of five different compound concentrations. For each measurement, the shutter was opened for 5 s, and the number of times the mouse licked the drinking spout was recorded. The shutter was then closed for 7.5 s, followed by presentation of the next taste solution. When testing hedonically negative compounds, we included additional one-second washout sessions with water between taste tests to cleanse the mouse's palate of the unpleasant compound and encourage it to keep licking the spout.

For each taste compound, we carried out separate statistical analyses. We averaged the results of identical tests for each compound and concentration to calculate the mean number of licks for each animal. Mixed-design analyses of variance with factors of group (wild type or experimental) and concentration were carried out using the values for each individual animal. Mice that failed to respond during any presentation of a particular concentration of a taste compound were excluded from statistical analyses for that compound. To determine differences between groups of mutant and control mice in consumption of specific taste solution concentrations, and to assess differences in the response of each group to individual taste compound concentrations, we used post hoc least-significant difference tests (STATISTICA10, Stat Soft Inc.). $P < 0.05$ was considered significant in all analyses.

**PLAs.** The following primary antibodies were used: mouse anti-β-catenin (Sigma-Aldrich Co., C-7207, 1:1,000); rabbit anti-β-catenin (Novus Biologicals, NBP1-32239, 1:200); goat anti-β-catenin (R&D Systems, AF-1329, 1:100); rabbit anti-KLF4 (Cell Signaling Technology, 4038S, 1:100); goat anti-KLF4 (R&D systems, AF-3158, 1:50); rabbit anti-LEF1 (Cell Signaling Technology, 2286, 1:100); rabbit anti-TCF3 (EMD Millipore, ABE414, 1:800; Cell Signaling Technology, 2883, 1:100); and anti-TCF4 (Cell Signaling Technology, 2565, 1:100). A Duolink PLA kit (Olink Bioscience, Uppsala, Sweden) was used according to the manufacturer's instructions with minor modifications. 4–6 μm frozen sections were fixed with 4% formaldehyde for 5 min, blocked in PBS/5% donkey serum/0.8% Triton X-100 for 2 h, incubated in primary antibodies overnight at 4 °C, washed in PBST, and incubated in PLA probe anti-goat plus or anti-mouse plus, and PLA probe anti-rabbit minus for 2 h. Ligation and amplification steps were performed following the manufacturer's protocol. The slides were washed twice in washing buffer and once in PBS for 20 min each. Slides were dried and mounted using Duolink mounting medium and photographed using a Leica Microsystems DM5500B fluorescent microscope. All assays were repeated with at least three biological replicates.

**ChIP assays.** ChIP assays were performed using the Chromatin Immunoprecipitation kit (Cell Signaling Technology, Danvers, MA). Mouse tongue or footpad epidermal layers were separated following Dispase II treatment, washed with PBS and cross-linked with 1% formaldehyde for 20 min. For β-catenin ChIP, tissues were cross-linked with ethylene glycol-bis(succinimidyl succinate) (Thermo Fisher Scientific) at 5 mM final concentration, with addition of formaldehyde to 1% final concentration after 30 min of incubation. All steps were performed at 4 °C unless otherwise indicated. Tissue was pulverized under liquid nitrogen with a mortar and pestle and lysed in ChIP buffer. Chromatin was sonicated to 200–900 bp using an S220 Focused-ultrasonicator (Covaris, Woburn, MA). Thirty microgram chromatin were incubated overnight at 4 °C with 10 μl anti-β-catenin antibody (Cell Signaling Technology, 9587P), anti-KLF4 (R&D Systems, AF-1329), anti-TCF3 (Active Motif, 61125) or control IgG, followed by addition of Protein G Magnetic Beads. Beads were rinsed in washing buffer and DNA analysed by qPCR. At least three biological replicates were analysed in each experiment, and three technical replicates were performed for each biological sample. Primers are listed in Supplementary Table 1.

**Data availability.** The authors declare that the data supporting the findings of this study are either available within the paper and its Supplementary Information files, or are available from the corresponding author on reasonable request.

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

## Acknowledgements

We thank members of the Millar and Morrisey labs for helpful discussions; our patients for their involvement; Deena Leslie Pedrioli for anti-mouse KRT9; Maria Morasso for anti-DLX3; Adam Glick for *Krt5-rtTA* mice; Makoto Taketo for *Ctnnb1*$^{(Ex3)fl/+}$ mice; Robert Granger and Catriona Brennan for *WNT10A*c.391G>A patient tissue; Ning Li for *R26R-rtTA tetO-H2B-GFP* tissue; Michael Passanante for patient photography; Alexander Wright for micro-CT; Tiffany Aleman and Hillary Ellis for assistance with taste-testing experiments; and PSOM Transgenic and CDB/CVI Imaging Core facilities for ES cell injection and SEM. NIH R37AR047709 (S.E.M.), RO1DE024570 (S.E.M.), R56DE023100 (S.E.M.), Penn Skin Biology and Diseases Resource-based Center (P30AR069589), Penn Skin Disease Research Center (P30AR057217), Monell Pheno-typing Core (P30DC011735) and Dermatology Foundation (M.X.) supported the study.

## Author contributions

M.X., J.H., E.Y.C. and S.E.M. designed the study with help from D.G., L.A.B., K.H.K., J.A.M., J.P.K. and E.E.M. M.X., J.H., M.S., J.C., H.G., C.M.S., F.L., C.M., J.E.D., B.J.C., M.T., F.L. and X.Z. performed experiments. S.K., B.L. and E.C. provided patient data. J.T.S. and A.I.R. interpreted data. M.X. and S.E.M. wrote the manuscript.
