## [Peer review file · Nature Communications]

Reviewers' comments:

Reviewer #1 (Remarks to the Author):

This is a carefully performed study addressing the function of one of the Wnt signals, Wnt10A. The focus is on organs and tissues developing as appendages of the epithelium, some of which were already known to depend on Wnt10a, in particular teeth and skin. The Wnt10A loss-of-function phenotypes were analysed both in human and mouse. Several mouse models were utilized, allowing for example inactivation of Wnt10a at various time points. The phenotypic analyses are remarkably detailed, and the phenotypes were examined during embryogenesis and followed postnatally and indicated contribution of Wnt10a to a variety of developmental processes.

The paper is very well written and presents exceptionally large amounts of high-quality and novel data. The careful analysis of multiple epithelial organs and tissues demonstrates that Wnt10A can have different cellular functions depending on the tissue. The new and interesting findings include the unexpected role of Wnt10A in cell differentiation, and the identification of transcription factor KLF as a specific tissue-specific partner of betacatenin is novel and interesting.

The authors have explored several possible mechanisms whereby loss of Wnt10a could have caused the phenotypes in different organ systems, particularly organs forming as epithelial appendages. This massive amount of work indicates that loss of Wnt10a affects proliferation of progenitor cells, and this part includes an extensive lineage tracing of Axin2 positive stem/progenitor cells and analysis of their contributions to the different organs and tissues. These are remarkably valuable data for numerous scientists. It was in addition discovered that some specific abnormalities resulted from the lack of Wnt10a in cell differentiation. These findings explain the pathogenesis of the Wnt10A mutant human and mouse phenotypes of the skin surface.

Finally, the authors focused on transcriptional partners of betacatenin and demonstrated that they depended on the tissue. This brings valuable data on the activation of distinct transcriptional programs by betacatenin signaling in different tissues.

This work has generated large amounts of new information on Wnt responding cells and the roles of Wnt signaling in several epithelial organs, and the authors have succeeded in presenting and discussing the data in a logical and understandable manner.

The exceptionally large scope and amounts of data may, however be a drawback. Reading the massive article is challenging, although it is well written. And particularly, many of the interesting findings may not reach all scientists who would be interested, for example in different organs, axin2 lineage tracings of progenitor cells, mouse and human analyses. Extending the abstract and the number of key-words could help in this problem.

This article will definitely interest many different audiences: from developmental biologists to human geneticists and clinicians. Wnt10A is associated with a variety of human diseases – from cancer to congenital defects, for example missing teeth and hereditary skin disorders.

Reviewer #2 (Remarks to the Author):

This manuscript contains innovative data on the role of Wnt10a in the control of skin development, skin appendage maintenance and regeneration. The authors convincingly demonstrate that Wnt10a serves as critical determinant controlling adult epithelial proliferation and region-specific differentiation. The authors show that human WNT10A mutations are associated with developmental tooth abnormalities and adolescent onset of a broad range of ectodermal defects. They correlate these findings with a number of genetically modified mouse models demonstrating that β -catenin pathway activity and adult epithelial progenitor proliferation are reduced in the absence of WNT10A, and identify Wnt-active self-renewing stem cells in the affected tissues. Finally, the authors identify Klf4 transcription factor co-partner interacting with β -catenin in differentiating, but not proliferating, cells to promote expression of specialized keratins including KRT9 required for normal tissue structure and integrity.

The manuscript is excellently written and illustrated. The data presented in the manuscript are clear and significantly advance our understanding of the role of Wnt signaling pathway in the control of organ development, regeneration and maintenance.

There only few minor issues for the authors to address:

1. Is Wnt10a expressed in the mesenchymal cells (hair follicle dermal papilla, sweat gland mesenchyme, etc.)?
2. What is known about upstream regulators controlling Wnt10 expression in epithelial cells?
3. Data on hair follicle miniaturization in adult mice upon Wnt10 ablation are very exciting and suggest an accelerated hair follicle ageing. Did the authors check the expression of Col17a1 expression that contribute to the hair follicle ageing (PMID: 26912707) in the Wnt10a-deficient hair follicles?

Reviewer #3 (Remarks to the Author):

While WNT10A mutations are a well-known cause of ectodermal dysplasia, and the role of Wnt10a in tooth morphogenesis, including in Wnt10a knockout mice, has been extensively studied, this manuscript contains a number of new, significant observations. In particular, the authors identified and characterized an interesting, novel role of Wnt10A in concert with KLF4 to control regionally restricted epidermal differentiation. From the perspective of skin biology, this manuscript offers significant insights into the interesting phenomena of distinct epidermal differentiation programs in plantar epidermis and the tongue. Furthermore, the tooth aspect of the manuscript adds much greater depth compared to previous studies, and a number of other ectodermal appendages are included in the characterization of the Wnt10a ko mice. The study also has translational relevance as it suggests a treatment target for the palmoplantar symptoms of patients with WNT10a-mutated ectodermal dysplasia. Overall, the data is of high quality; this is a very strong paper that offers significant advances.

Specific comments:

1. The nature of the hair abnormality in the patient and in Wnt10a mutated mice is somewhat unclear. Given the adult onset, the authors seem to be arguing that this is a stem cell function-related phenotype. However, this is not so clear to this reviewer. Are there any hair shaft abnormalities in the patient (and in other WNT10A mutated patients)? Hair loss can be caused by fragile hair shafts which relate more to the proliferation and differentiation of the hair shaft than stem cell function. The hair phenotype observed when Wnt10a was deleted at P9 seems more consistent with abnormalities in the proliferation and differentiation of the hair shaft than stem cell abnormality. In this respect, the hair phenotype may in part be quite analogous to the region-specific differentiation phenotype in the interfollicular epidermis. The authors should clarify the discussion of this aspect of the study.
2. The Axin2Cre-based strategy for identifying Wnt-targeted stem cells is a bit concerning due to the potential leakiness of this marker, and the question of whether axin2 expression is only dependent on Wnt activation.
3. The expression of Wnt10a is described in Suppl. Fig. 1o-t. It would be useful to clarify whether the Wnt10a expression is restricted to plantar and footpad epidermis. Is there a sharp expression boundary between plantar and non-plantar epidermis?
4. A previous report on Wnt10a ko mice described supplemental teeth in contrast to missing teeth in patients with Wnt10a mutations. The authors should comment on this.
5. Several of the OODD cases have been described to have hyperhidrosis of palms and soles. Here the authors state that their patient has decreased sweating (it is difficult to fully understand Fig. 1e,f without further explanations, but superficially the image seems to show that the patient has greater sweating, contradicting the authors' statement). It would be useful if the authors commented on their data in this respect, and how to reconcile increased sweating with the stem cell phenotype and abnormal sweat glands with impaired sweating in ko mice.

6. The authors claim an effect on region-specific terminal differentiation in the epidermis. Krt9 is not a terminal differentiation marker and therefore it would be good to show an effect (if it there) on more canonical markers such as Tgm, Lor, Flg etc.

7. In Fig. 4g and h, the authors state that there is a decreased expression of taste bud markers KRT8 and SOX2 in sectioned circumvallate papillae. This data is not crystal clear and it would be good to provide some quantification.

Dual requirements for WNT10A in proliferation and KLF4-mediated differentiation underlie ectodermal dysplasia

Response to Reviewers

We have shortened the text and Figure Legends, and moved as much of the previous Supplementary Data as possible into the main Figures, in order to comply with the format requirements for *Nature Communications*. As these changes were substantial, we have not highlighted them. However, we have highlighted changes in the text that were made in response to the reviewers' suggestions.

Reviewer #1

This work has generated large amounts of new information on Wnt responding cells and the roles of Wnt signaling in several epithelial organs, and the authors have succeeded in presenting and discussing the data in a logical and understandable manner. The exceptionally large scope and amounts of data may, however be a drawback. Reading the massive article is challenging, although it is well written. And particularly, many of the interesting findings may not reach all scientists who would be interested, for example in different organs, axin2 lineage tracings of progenitor cells, mouse and human analyses. Extending the abstract and the number of key-words could help in this problem.

We thank the reviewer for his/her very positive comments regarding the scope and significance of this work. To help ensure that our findings reach interested scientists we have edited the abstract and included more key words, as suggested.

Reviewer #2

This manuscript contains innovative data on the role of Wnt10a in the control of skin development, skin appendage maintenance and regeneration.... The data presented in the manuscript are clear and significantly advance our understanding of the role of Wnt signaling pathway in the control of organ development, regeneration and maintenance.... There only few minor issues for the authors to address.

We are grateful for the reviewer's enthusiasm for this work and have addressed the minor issues raised:

1. Is Wnt10a expressed in the mesenchymal cells (hair follicle dermal papilla, sweat gland mesenchyme, etc.)?

We mention that *Wnt10a* is expressed in hair follicle (HF) dermal papilla (DP) as well as in HF epithelial cells. In response to Reviewer 3 (comment 3) we now also point out that *Wnt10a* is expressed at similar levels in hair skin epidermis, plantar epidermis and footpad epidermis. We detected *Wnt10a* expression in sweat gland myoepithelial cells, but not in sweat gland mesenchyme, by in situ hybridization. The revised text is on **page 6, lines 135-141**:

*"Wnt10a expression coincides with β -catenin signaling in molar tooth development^{20,21}, in embryonic anlagen for HFs and taste papillae, in adult interfollicular epidermis, and in HF epithelial cells and DP^{16,22-24}. We detected *Wnt10a* expression in plantar and footpad epidermis at similar levels to those in haired skin epidermis, and in regenerating adult*

epithelia including filiform and fungiform papillae and sweat ducts (**Fig. 2o-t**). *Wnt10a* localized to sweat gland myoepithelial cells, but was not detectable in sweat gland mesenchyme (**Fig. 2p**).”

2. What is known about upstream regulators controlling Wnt10 expression in epithelial cells?

We thank the reviewer for raising this question. Our previously published data suggest that upregulated *Wnt10a* expression in embryonic hair follicle placodes requires NF- κ B signaling (Zhang et al. Dev Cell. 2009 Jul;17(1):49-61); however this is unlikely to account for more generalized epithelial *Wnt10a* expression. Deletion of epidermal Wntless causes reduced *Wnt10a* expression in skin epithelia, suggesting that expression of *Wnt10a* may be regulated directly or indirectly by epithelial Wnt ligands (Fu J, Hsu W. J Invest Dermatol. 2013 Apr;133(4):890-8). The clock gene *Bmal1* directly regulates *Wnt10a* expression in pre-adipocytes (Guo et al. FASEB J. 2012 Aug;26(8):3453-63), but it is unknown whether this is also the case in epithelial cells. We agree with the reviewer that regulation of *Wnt10a* expression in diverse epithelia is an important area for future study; however we feel that it is beyond the scope of this manuscript.

3. Data on hair follicle miniaturization in adult mice upon Wnt10 ablation are very exciting and suggest an accelerated hair follicle ageing. Did the authors check the expression of Col17a1 expression that contribute to the hair follicle ageing (PMID: 26912707) in the Wnt10a-deficient hair follicles?

We appreciate this interesting suggestion. As the reviewer points out, some hair follicles in normal mice become miniaturized and lose stem cell markers at 18-34 months of age via a mechanism involving proteolysis of COL17A1. By contrast, in *Wnt10a* mutants we observe hair follicle miniaturization by 6 months of age. In response to the reviewer's suggestion we performed immunofluorescence experiments to check whether expression of COL17A1 is altered in miniaturized hair follicles from 6 month-old *Wnt10a* mutants. Our results indicated that, unlike in aging, COL17A1 expression is maintained in miniaturized *Wnt10a* mutant hair follicles. This result is consistent with our finding that these follicles maintain expression of the stem cell markers KRT15 and CD34, and suggests that the *Wnt10a* mutant miniaturization phenotype is not due to accelerated aging. Consistent with this, levels of *Axin2* expression are reported to be similar in young and aged hair follicles (Matsumura et al. Science. 2016 Feb 5;351, 575). These new data are included in **Supplementary Figure 2**, panels (s,t) and are described on **page 10, lines 220-224**:

“In normal aged mice (18-34 months), hair follicles miniaturize via loss of stem cells due to COL17A1 proteolysis³³. Miniaturized HFs of 6-month old *Wnt10a* mutants expressed COL17A1 (**Supplementary Fig. 2s,t**), consistent with maintenance of stem cell markers and suggesting that miniaturization was not caused by accelerated aging. In line with this, levels of *Axin2* expression are similar in young and aged hair follicles³³.”

We have also revised the Methods section to include immunofluorescence with anti-COL17A1 (**page 29, line 710**).

Reviewer #3

Overall, the data is of high quality; this is a very strong paper that offers significant advances.

We are grateful to the reviewer for this positive assessment of our manuscript, and have responded to each point raised in detail below.

1. The nature of the hair abnormality in the patient and in Wnt10a mutated mice is somewhat unclear. Given the adult onset, the authors seem to be arguing that this is a stem cell function-related phenotype. However, this is not so clear to this reviewer. Are there any hair shaft abnormalities in the patient (and in other WNT10A mutated patients)? Hair loss can be caused by fragile hair shafts which relate more to the proliferation and differentiation of the hair shaft than stem cell function. The hair phenotype observed when Wnt10a was deleted at P9 seems more consistent with abnormalities in the proliferation and differentiation of the hair shaft than stem cell abnormality. In this respect, the hair phenotype may in part be quite analogous to the region-specific differentiation phenotype in the interfollicular epidermis. The authors should clarify the discussion of this aspect of the study.

We agree with the reviewer that Wnt10a performs multiple functions in hair follicles. As described in the manuscript, *Wnt10a* deletion at P9 causes premature catagen (**Figure 5a,b**) and is associated with decreased proliferation of matrix progenitor cells (**Figure 5a',b'**). *Wnt10a* deletion at P18 causes significantly delayed anagen onset (**Figure 5c,d**), which is caused by failure of timely proliferation of secondary hair germ cells (**Figure 5c',d'**). We now explain this more fully (**page 9, lines 198-203**):

*“Wnt10a loss from P9 (embryonic anagen) caused premature HF regression, cessation of matrix cell proliferation, and decreased cyclin D1 expression (**Fig. 5a-b',j; Supplementary Fig. 2k,l**). Deletion at P18 delayed initiation of anagen, indicated by histology and absent SHG proliferation (**Fig. 5c-d'**), and prevented timely *TL-GFP* activation and external hair growth (**Fig. 5k-m**).”*

We agree with the reviewer that *Wnt10a* loss may also cause defects in hair shaft differentiation. To address this issue, we examined expression of hair shaft keratins in mutant and control anagen hair follicles at P33 using immunofluorescence for the pan-hair shaft keratin antibody AE13. Our data show that hair shaft keratins are expressed in the mutant follicles (new **Supplementary Figure 2a,b**). We also examined hair shaft cuticle structure by scanning EM, and found that this appeared similar in mutant and control mice, although the mutant hair shafts had a smoother appearance that was likely due to excess sebum production (new **Supplementary Figure 2c,d**). As expected from the reduced matrix cell proliferation, mutant hair shafts were shorter and thinner than controls (new **Supplementary Figure 2e,f**). High magnification light microscopy showed that internal hair shaft structures were disorganized in mutants, consistent with the reviewer's suggestion that some aspects of differentiation might be disturbed (new **Supplementary Figure 2g-j**). We describe these data on **page 9, lines 193 – 197**:

*“Immunofluorescence for pan-hair shaft keratins produced a signal in mutant hair follicles and cuticle structure appeared grossly normal by scanning EM (SEM); however mutant hair shafts were shorter and thinner than controls, with disorganized internal structures (**Supplementary Fig. 2a-j**).”*

Differentiation of the hair shaft is complex, requiring expression of many different hair

keratins and keratin associated proteins, and producing multiple layers including the medulla, cortex and cuticle. In the current study we therefore chose to analyze the effects of *Wnt10a* signaling in differentiation by studying two much simpler structures, the tongue filiform papilla and plantar epidermis. However, we agree that similar principles may apply to the hair shaft. We now discuss this on **page 23, lines 548-553**:

“This mechanism may be generalizable to other specialized epithelia such as HFs, TBs, sweat ducts and nails, that exhibit β -catenin signaling in differentiating as well as proliferating cells. In line with this, in addition to controlling HF progenitor cell proliferation⁶¹, Wnt/ β -catenin signaling has been implicated in the regulation of hair keratin gene expression^{62,63}, and we observed disorganized internal hair shaft structures in *Wnt10a* mutant mice.”

We have also revised the Methods section to include immunofluorescence with AE13 (**page 29, lines 708-709**).

2. The Axin2Cre-based strategy for identifying Wnt-targeted stem cells is a bit concerning due to the potential leakiness of this marker, and the question of whether axin2 expression is only dependent on Wnt activation.

We analyzed un-induced tissues to carefully control for leakiness of the Axin2-Cre in combination with the different Cre reporter lines used (**page 14, lines 320-324; Supplementary Figure 5c**) and by analyzing data from multiple samples. However, we agree with the reviewer that, as in any Cre-based lineage tracing method, we cannot entirely exclude the possibility that some marked clones arose from leaky Cre activity. We have added this caveat in the Results section (**page 14, lines 324-326**):

“Un-induced *Axin2-Cre*^{ERT2tdT} *R26R*^{EYFP} mice displayed sporadic EYFP labeling in palatal rugae and tongue muscle, but not in skin or tongue epithelia (**Supplementary Fig. 5c** and data not shown). We did not detect leakiness in any of these tissues when *Axin2-Cre*^{ERT2tdT} was used with *R26R*^{mTmG} or *R26R*^{Confetti} (data not shown); however, as in any lineage tracing experiment, we cannot absolutely exclude that some clones resulted from leaky Cre activity.”

We agree with the reviewer that we cannot exclude the possibility that additional pathways regulate *Axin2*, and we now include this caveat in the Discussion (**page 20, lines 483-488**). However, it is important to note that previous studies (e.g. Lim et al. Proc Natl Acad Sci U S A. 2016 Mar 15;113(11):E1498-505) have shown that deletion of epithelial β -catenin causes loss of *Axin2* expression:

“Using lineage tracing experiments with *Axin2-Cre*^{ERT2}, we identified self-renewing *Axin2*-expressing cells in the tissues affected by loss of WNT10A function. While we cannot exclude that additional pathways regulate *Axin2*, its expression in skin epithelia is removed by β -catenin deletion⁴². Thus, our data suggest that Wnt-active self-renewing stem cells contribute to regeneration of *Wnt10a*-dependent tissues.”

3. The expression of Wnt10a is described in Suppl. Fig. 1o-t. It would be useful to clarify whether the Wnt10a expression is restricted to plantar and footpad epidermis. Is there a sharp expression boundary between plantar and non-plantar epidermis?

We now clarify that *Wnt10a* is expressed at similar levels in interfollicular, plantar and footpad epidermis (**page 6, lines 137-138**). Please also see response to Reviewer 2, comment 1.

4. A previous report on *Wnt10a* ko mice described supplemental teeth in contrast to missing teeth in patients with *Wnt10a* mutations. The authors should comment on this.

We re-examined our global *Wnt10a*^{-/-} mice and found that, consistent with the previous report by Yang et al. (Mol Genet Genomic Med. 2015 Jan;3(1):40-58), approximately 50% of these mutants develop a loosely anchored ectopic M4 molar (this ectopic tooth is generally lost during processing for micro-CT). We now include these data in new **Supplementary Figure 1a,b**. We also observed ectopic molar M4 development in a subset of constitutive epithelial-specific *Wnt10a* mutants, indicating that epithelial *Wnt10a* suppresses formation of ectopic M4 molars. β -catenin signaling in incisor region mesenchyme prevents ectopic incisor development by stimulating expression of *Bmp4* and *Shh*, which then act to limit Wnt activity (Fujimori et al, Dev Biol. 2010 Dec 1;348(1):97-106). *Wnt10a* may be involved in a similar feedback mechanism to suppress ectopic molar formation by signaling to molar mesenchyme. In line with this possibility, *Wnt10a*^{-/-} mandibles displayed reduced expression of *Shh* at E14.5 (**Fig. 3f**). These data are now described and discussed on **page 7, lines 149-166**:

“Approximately 50% of *Wnt10a*^{-/-} mice developed loosely anchored ectopic molar M4 teeth (**Supplementary Fig. 1a,b**). Maxillary and mandibular molar teeth had flattened cusps, reduced size, and defective root bifurcation and extension compared with littermate controls (**Supplementary Fig. 1a,b; Fig. 3a-c**), consistent with a previous report²⁵. Cusp and size abnormalities were observed by E17.5 (**Fig. 3d**), indicating their morphogenetic origin, and mimicked the effects of Wnt/ β -catenin inhibition after tooth initiation²⁰. By one year of age, alveolar bone density was markedly decreased in *Wnt10a*^{-/-} mice compared with controls, in line with human patient phenotypes (**Fig. 1k**), and molar teeth were frequently missing (**Fig. 3b,c**). *Krt14-Cre Wnt10a*^{fl/fl} mutants displayed molar cusp and root defects, but not decreased bone density (**Fig. 3e,j**), and a subset of these mice formed ectopic M4 molars (not shown). Thus, epithelial *Wnt10a* is required for normal tooth morphogenesis and suppression of ectopic molar formation, and mesenchymal *Wnt10a* maintains alveolar bone. β -catenin signaling in incisor region mesenchyme prevents ectopic incisor development by stimulating expression of *Bmp4* and *Shh*, which then act to limit Wnt activity²⁶. *Wnt10a* may be involved in a similar feedback mechanism to suppress ectopic molar formation by signaling to molar mesenchyme. In line with this possibility, *Wnt10a*^{-/-} mandibles displayed reduced expression of *Shh* at E14.5 (**Fig. 3f**).”

As the reviewer points out, patients with *WNT10A* mutations have missing teeth; however, it should be noted that tooth agenesis affects the secondary, not the primary dentition in these patients, and mice do not develop a secondary dentition. We now discuss this on **page 22, lines 526-529**:

“While microdontia and taurodontism are concordant between human patients and mouse mutants, patients with *WNT10A* mutations also display secondary tooth agenesis. Lack of this phenotype in mouse mutants is not surprising, as mice do not develop a secondary dentition.”

As the reviewer also notes, ectopic primary molars develop in *Wnt10a* mutant mice but have not been described in human patients with *WNT10A* mutations. We discuss possible reasons for this discrepancy on **page 22, lines 529-535**:

“Interestingly, some *Wnt10a* mutant mice develop ectopic molar M4 teeth that are loosely anchored and preferentially lost, a phenotype that has not been described in human *WNT10A* patients. Careful examination of the dentition in very young, severely affected patients might similarly reveal ectopic formation of primary molars. Alternatively, differences in the mechanisms controlling patterning of dental development in humans and mice could explain the absence of ectopic primary molar formation in human patients.”

5. Several of the OODD cases have been described to have hyperhidrosis of palms and soles. Here the authors state that their patient has decreased sweating (it is difficult to fully understand Fig. 1e,f without further explanations, but superficially the image seems to show that the patient has greater sweating, contradicting the authors' statement). It would be useful if the authors commented on their data in this respect, and how to reconcile increased sweating with the stem cell phenotype and abnormal sweat glands with impaired sweating in ko mice.

We apologize for the poor quality of the photographs provided in the original version of **Figure 1e,f**. We have now included much higher magnification photos, which clearly show that, similar to several other human *WNT10A* pedigrees (e.g. Tziotzios et al. 2014 Br J Dermatol 171, 1211-4; Zeng et al. 2016. Genes 7, 65), our *WNT10A* patient has decreased sweating on her palms, consistent with the phenotype observed in *Wnt10a* mutant mice. As the reviewer points out, the opposite phenotype has been described in other *WNT10A* human patients (e.g. Tziotzios et al. 2014 Br J Dermatol 171, 1211-4; Adaimy et al. 2007. Am J. Human Genet. 81, 821-8). We speculate that compensatory mechanisms in some patients, for instance upregulation of other Wnt ligands during epithelial development, could account for these apparently contradictory findings. We now include discussion of this issue on **page 10, lines 232-240**:

“Sweat gland ducts failed to extend in *Wnt10a*^{-/-} mutants (**Fig. 6r,s**), or following postnatal epithelial β -catenin deletion (**Supplementary Fig. 3m,n**), and starch-iodine tests revealed a functional inability to sweat (**Supplementary Fig. 3k-l'**). Thus, *WNT10A*/ β -catenin signaling is required to complete postnatal oral appendage and sweat duct development. In line with this, our patient, and several other human *WNT10A* pedigrees^{5,34}, display palmoplantar hypohidrosis. However, palmoplantar hyperhidrosis has also been described in some *WNT10A* patients^{5,35}. We speculate that variable compensatory mechanisms, for instance upregulation of other Wnt genes during development, could account for these disparate findings.”

6. The authors claim an effect on region-specific terminal differentiation in the epidermis. Krt9 is not a terminal differentiation marker and therefore it would be good to show an effect (if it there) on more canonical markers such as Tgm, Lor, Flg etc.

We thank the reviewer for pointing out that KRT9 is a suprabasal differentiation protein, not a marker of terminal differentiation. We have now corrected this mistake throughout the text. As suggested by the reviewer we have now examined the expression of the terminal differentiation markers filaggrin and involucrin in footpad epidermis of *Wnt10a*^{-/-} mice, and following inducible deletion of epithelial β -catenin. Consistent with the regional

specificity of the skin phenotypes seen in mice and humans, expression of these general terminal differentiation proteins was not affected by loss of *Wnt10a* or β -catenin (new **Supplementary Figure 6, panels (k-r)**). In line with this, expression of the terminal differentiation protein loricrin was similar in control and WNT10A patient plantar skin (new **Supplementary Figure 6, panels (s,t)**). These data are now described on **page 17, lines 402-411**:

“Global *Wnt10a* loss or inducible epithelial β -catenin deletion caused decreased KRT9 protein and mRNA levels (**Fig. 9n-r**). By contrast, the suprabasal keratin gene *Krt10* and the terminal differentiation proteins filaggrin and involucrin, which locate to epidermis in all body regions, were unaffected by *Wnt10a* or β -catenin deletion (**Supplementary Fig. 9j-r**). Similarly, while KRT10 and loricrin expression was comparable in WNT10A patient and sex- and age-matched control plantar epidermis (**Fig. 9s'-t'**, **u'-v'**; **Supplementary Fig. 6s,t**), KRT9 protein was decreased in our *WNT10A c.756+1 G>A* patient, and in an unrelated patient homozygous for a *WNT10A c.391 G>A* mutation⁴ (**Fig. 9s-v'**), and *KRT9* mRNA levels were lower in patient plantar skin than in control (**Supplementary Fig. 6u**).”

We have also revised the Methods section to include immunofluorescence with antibodies to filaggrin, involucrin and loricrin (**page 29, line 710-711**).

7. In Fig. 4g and h, the authors state that there is a decreased expression of taste bud markers KRT8 and SOX2 in sectioned circumvallate papillae. This data is not crystal clear and it would be good to provide some quantification.

As SOX2 is expressed in basal taste bud progenitors as well as in differentiated Type I taste bud cells, we chose to quantify the data for expression of KRT8 in circumvallate papillae. As circumvallate papillae are generally smaller in *Wnt10a* mutant mice than in controls, we quantified the percentage of DAPI+ cells that were positive for KRT8 within each papilla structure. These data showed that the percentage of differentiated KRT8+ cells is lower per mutant papilla than per control papilla. These data are now included in revised **Figure 6, panel (i)** and are described in the text on **page 11, lines 251-253**. Methods used for quantitation are described in the Methods section, **page 28, lines 677-681**.

Additional Revisions

1. In addition to the revisions requested by the reviewers we have now included more comprehensive results from taste testing of additional *Wnt10a*^{-/-} and littermate control mice (revised **Supplementary Figure 4**). We now include results from taste testing using two-bottle choice tests as well as brief-access gustometer tests. The results of these experiments did not reveal significant differences in the taste responses of *Wnt10a*^{-/-} mice compared with littermate controls, consistent with the data presented in the original version of our paper. We did note a slightly higher intake of water in mutants that had been subjected to mild restriction of food and water intake prior to testing for sweet flavor discrimination. This difference was not observed in mice that had been subjected to more severe food and water-restriction e.g. before testing for sour, bitter, HCl, etc. We speculate that a slight barrier defect in the mutant mice could cause increased dehydration relative to controls, which might be revealed after mild water/food

restriction but would be masked by the effects of more severe restriction which causes all of the mice to take in increased water. The revised text is on **page 11, lines 258-264**:

“However, we did not detect significant differences in the taste responses of adult mutants versus controls for sweet, sour, salt, or bitter tastes, or for the irritant, capsaicin (**Supplementary Fig. 4**), indicating that, as in our human patient, residual taste function is sufficient to discriminate these compounds. Interestingly, *Wnt10a*^{-/-} mutants had higher water intakes than controls following mild, but not more severe, water-restriction. This could reflect increased dehydration, possibly caused by a slight epidermal barrier defect.”

The Methods section has also been revised to include two-bottle choice tests and additional mice tested (**pages 31-32, lines 760-772**). Detailed procedures are provided in Supplementary Methods.

2. We have also replaced the data on SNAP-25 expression in *Axin2* lineage tracing experiments (previous **Supplementary Figure 6, panels l,m**) with higher quality images of circumvallate papilla (new **Figure 8, panels i,j**).

REVIEWERS' COMMENTS:

Reviewer #2 (Remarks to the Author):

No comments

Reviewer #3 (Remarks to the Author):

The authors have fully addressed my comments. I have no further comments on this interesting, well-conducted study.

NCOMMS-16-21617-A: Response to Referees

The reviewers had no further comments on our manuscript. We thank them for their careful reviews and positive evaluation of this study.

Reviewer #2 (Remarks to the Author):

No comments

Reviewer #3 (Remarks to the Author):

The authors have fully addressed my comments. I have no further comments on this interesting, well-conducted study.